# Deciphering the actin structure-dependent preferential cooperative binding of cofilin

**Kien Xuan Ngo[1]\*[†], Huong T Vu[2†], Kenichi Umeda[1], Minh-Nhat Trinh[3], Noriyuki Kodera[1], Taro Uyeda[4]**

[1]Nano Life Science Institute (WPI-NanoLSI), Kanazawa University, Kanazawa, Japan; [2]Centre for Mechanochemical Cell Biology, Warwick Medical School, Coventry, United Kingdom; [3]School of Electrical and Electronic Engineering, Hanoi University of Science and Technology, Hanoi, Viet Nam; [4]Department of Physics, Faculty of Advanced Science and Engineering, Waseda University, Shinjuku, Tokyo, Japan

**Abstract** The mechanism underlying the preferential and cooperative binding of cofilin and the expansion of clusters toward the pointed-end side of actin filaments remains poorly understood. To address this, we conducted a principal component analysis based on available filamentous actin (F-actin) and C-actin (cofilins were excluded from cofilactin) structures and compared to monomeric G-actin. The results strongly suggest that C-actin, rather than F-ADP-actin, represented the favourable structure for binding preference of cofilin. High-speed atomic force microscopy explored that the shortened bare half helix adjacent to the cofilin clusters on the pointed end side included fewer actin protomers than normal helices. The mean axial distance (MAD) between two adjacent actin protomers along the same long-pitch strand within shortened bare half helices was longer (5.0–6.3 nm) than the MAD within typical helices (4.3–5.6 nm). The inhibition of torsional motion during helical twisting, achieved through stronger attachment to the lipid membrane, led to more pronounced inhibition of cofilin binding and cluster formation than the presence of inorganic phosphate (Pi) in solution. F-ADP-actin exhibited more naturally supertwisted half helices than F-ADP.Pi-actin, explaining how Pi inhibits cofilin binding to F-actin with variable helical twists. We propose that protomers within the shorter bare helical twists, either influenced by thermal fluctuation or induced allosterically by cofilin clusters, exhibit characteristics of C-actin-like structures with an elongated MAD, leading to preferential and cooperative binding of cofilin.

## eLife assessment

In this manuscript the authors present high-speed atomic force microscopy (HSAFM) to analyze real-time structural changes in actin filaments induced by cofilin binding. This **important** study enhances our understanding of actin dynamics which plays a crucial role in a broad spectrum of cellular activities based on **solid** experimental evidence. Some technical questions, however, remain, making the data interpretation **incomplete**.

## Introduction

Cofilin is a member of the actin-depolymerizing factor/cofilin (ADF/cofilin) family that is present in all eukaryotes. There are two major isoforms of cofilin in mammals: cofilin 1 and cofilin 2, predominantly found in non-muscle and muscle tissues, respectively (*Bamburg and Bernstein, 2010*). Cofilin is widely recognized as a key regulator of actin filament dynamics, particularly in nonequilibrium assembly

\*For correspondence:
kienosakaphd@gmail.com

[†]These authors contributed equally to this work

**Competing interest:** The authors declare that no competing interests exist.

and disassembly processes. Cofilin promotes the depolymerization and severing of actin filaments in a concentration-dependent manner (*Andrianantoandro and Pollard, 2006*) and collaborates or competes with other actin binding proteins (ABPs) in this process (*Bibeau et al., 2021*; *Grintsevich et al., 2016*; *Lappalainen and Drubin, 1997*; *Ngo et al., 2016*).

The structure of an actin filament has traditionally been represented as a simplistic atomic model derived from the G-actin structure in the Holmes model (*Holmes et al., 1990*) without considering the conformational changes occurring within each protomer during actin polymerization (*Kasai et al., 1962*; *Rich and Estes, 1976*). The rigid-body rotation of the G-actin crystal structure used in the Holmes model prompted the development of accurate models that consider structural changes within protomers, such as the X-ray diffraction Lorenz model (*Lorenz et al., 1993*) and Oda model (*Oda et al., 2009*).

The actin protomer consists of two domains, the inner domain (ID) and the outer domain (OD), with relatively limited contact between them. The polypeptide chain traverses between these domains twice: at the loop centered around residue Lys336 and at the linker helix Gln137-Ser145, which functions as the axis of a hinge between the domains. As a result, two clefts are formed between these domains (*Dominguez and Holmes, 2011*). The upper cleft, located between subdomain (SD2) and SD4, serves as a binding site for nucleotides and divalent ions like $Mg^{2+}$. On the other hand, the lower cleft, located between SD1 and SD3, is characterized by residues Tyr143, Ala144, Gly146, Thr148, Gly168, Ile341, Ile345, Leu346, Leu349, Thr351, and Met355, which are predominantly hydrophobic. This cleft serves as the primary binding site for most ABPs including cofilin and is also responsible for critical longitudinal contacts between actin protomers within the filament (*Dominguez, 2004*; *Fujii et al., 2010*; *Oda et al., 2009*).

The actin filament is a double-stranded helical structure formed by the crossover of two long-pitch helices; the distance between the crossover points is known as the half helical pitch (HHP), which is approximately 36 nm (*Fujii et al., 2010*; *Holmes et al., 1990*; *Lorenz et al., 1993*; *Oda et al., 2009*). A canonical half helix is composed of 13 actin protomers (6.5 protomer pairs). Thus, the mean axial distance (MAD) between two adjacent protomers along the same long-pitch strand is 5.5 nm [36 nm/(13/2)] (*Egelman, 1994*). However, long actin filaments exhibit variable helical twists (*Egelman et al., 1982*). Recent cryo-EM studies have demonstrated the atomic structures of short actin segments with cofilin (referred to as cofilactin; *Galkin et al., 2011*; *Huehn et al., 2020*; *Tanaka et al., 2018*). When cofilin binds to actin in a 1:1 ratio to form a cofilactin segment, the helices became supertwisted, and the HHP is decreased by approximately 25% (*McGough et al., 1997*), reducing the number of actin protomers to ~11 in each half helix (5.5 protomer pairs) (PDB 3J0S; *Galkin et al., 2011*). In a simple model, assuming that the MAD remains unchanged in both bare actin and cofilactin segments, the difference in the number of actin protomers impacts the formation of short and long HHPs. Nonetheless, whether the alterations in MAD dynamics are comparable between normal and supertwisted half helices remains uncertain. Therefore, additional research is needed to investigate this particular issue.

In this study, we utilized several techniques, including principal component analysis (PCA) and high-speed atomic force microscopy (HS-AFM). We aimed to understand how the allosteric conformational changes induced by cofilin clusters in neighboring bare actin filaments contribute to the preferential binding of additional cofilin molecules toward the pointed-end (PE) side, as reported in our previous study (*Ngo et al., 2015*). Therefore, we generated pseudo AFM images representing long bare actin filaments (referred to as F-actin structures) and cofilactin filaments without cofilin molecules (referred to as C-actin structures) using available protein data bank (PDB) structures such as 6VAU (bare actin filament from a partially cofilin-decorated sample) and 6VAO (human cofilin-1 decorated actin filament; *Huehn et al., 2020*). We measured the HHPs and the number of actin protomers per HHP in pseudo-AFM images and used them as the reference values for the data obtained from our real HS-AFM analyses. In principle, we classify that filament regions with supertwisted half helices and a reduced number of protomer pairs per HHP, as detected in our AFM measurements, are indicative of C-actin-like structure. We aimed to address several issues: (**i**) Does the number of actin protomers per HHP vary between typical F-actin and C-actin-like structure regions? (**ii**) Does the MAD change, and do these changes affect cofilin binding within the unbound actin regions adjacent to cofilin clusters on the PE and barbed-end (BE) sides? Are these changes quantitatively similar or distinct? (**iii**) Can we evaluate the impact of Pi on torsional flexibility during helical twisting of actin filaments to elucidate how they influence the preferential cooperative binding and expansion of cofilin clusters?

An innovative and previously unexplored facet of this study is the visualization of the actin protomer's structure within a bare half helix adjacent to small cofilin clusters on the PE and BE sides using HS-AFM. Our findings elucidate the dynamic structural changes in half helices and the number of actin protomers per HHP in normal F-actin. Our results deviated from the conventional ensemble-averaged helical twists usually observed in EM and X-ray diffraction, depicting a rigid half helix. Moreover, we show that shortened bare half helices adjacent to the cofilin clusters on the PE side contain less actin protomers than conventional half helices. The MAD within the shortened bare HHP (C-actin-like structure) is longer than the MAD within the canonical HHP (F-actin structure). This elongated MAD can be interpreted as resulting from the allosteric transmission of structural changes originating in the cofilactin region, extending beyond the boundary between cofilactin and bare actin regions. This extended influence affects the structure of the shortened bare half helix on the PE side. Additionally, our findings suggest that the cooperative binding of cofilin is significantly impacted by the torsional flexibility of actin filaments. The inhibition of torsional motion during helical twisting, achieved via physical perturbation, inhibited cofilin binding and the expansion of larger clusters more than Pi binding.

We also verify our previous hypothesis, which supports cofilin's initial binding to the favorable structure of actin protomers because F-actin, without ADF/cofilin proteins, could exist in the 162° twist state observed for cofilin-decorated F-actin (*Galkin et al., 2011*; *Galkin et al., 2001*; *Ngo et al., 2016*; *Ngo et al., 2015*). Our hypothesis focuses on the evidence of decreasing the number of protomers within the bare half helix adjacent to the cofilin cluster on the PE side. This reduction leads to a shorter HHP but paradoxically results in a longer MAD in the C-actin-like structure area. The protomers within a shortened bare half helix near cofilin clusters are converted to C-actin-like structures via our proposed F-actin to C-actin transition pathway, thereby facilitating the expansion of clusters on the PE side. If cofilin molecules fail to promptly attach to protomers within the shortened bare region, the protomers resembling the C-actin structure in that region can quickly change the closed nucleotide binding clefts to the open state and transition to a G-actin state. This transition (C-actin-like to monomer G-actin-like structure) ultimately results in severing of the filament at or near the boundary of cofilactin and bare actin regions. Furthermore, this study suggests that Pi binding mainly inhibits the initial binding of cofilin by reducing the fraction of supertwisted half helices in F-ADP.Pi-actin, whereas the removal of Pi enhances the initial binding of cofilin by elevating the fraction of supertwisted half helices in F-ADP-actin. Nevertheless, Pi binding only slightly inhibits the propagation of the supertwisted helical structure within cofilin clusters to bare half helices on the PE side in F-ADP.Pi-actin when compared to F-ADP-actin. Together, this clarifies that the initial interactions of cofilin with F-actin and the subsequent expansion of clusters are closely linked to the variability in helical twists.

## Results

### Principal component analysis suggests C-actin-like structure as the optimal actin structure for cofilin binding

To investigate the relationship between actin structure and cofilin binding, we explored the collective changes in actin conformations as they transitioned from G-actin to F-actin and then to C-actin. We applied PCA to identify the principal components (PCs), as depicted in *Figure 1*. Our analysis included an array of actin chains sourced from 41 available PDB structures of filamentous actin protomers bound to cofilin, profilin, DNAse I, and various nucleotides (ATP, AMPPNP, ADP.Pi, ADP, and Apo). We then compared these findings with five additional PDB structures featuring free G-actin and twinfilin-1-bound G-actin (*Table 1*).

We performed PCA with the same dataset using different alignments for subdomains 1 (SD1), 2 (SD2), 3 (SD3), and 4 (SD4), the outer domain (OD, SD1 + SD2), the inner domain (ID, SD3 + SD4), and the entire actin structure. In all cases, we observed clustering of different groups of actin chains. Interestingly, aligning only the IDs of all actin chains yielded the most meaningful PCs. Importantly, PCA can elucidate the conformational transition between C-actin and G-actin structures, and vice versa, through PC1, while PC2 delineates the pathway between F-actin and C-actin structures (*Figure 1A*). To visualize the conformational changes associated with each PC, we back-calculated and depicted the PCs in *Figure 1B and C*, respectively. These illustrations revealed two outer domain motion modes. The first mode (PC1, accounting for 45% of the structural variance) involved rotations

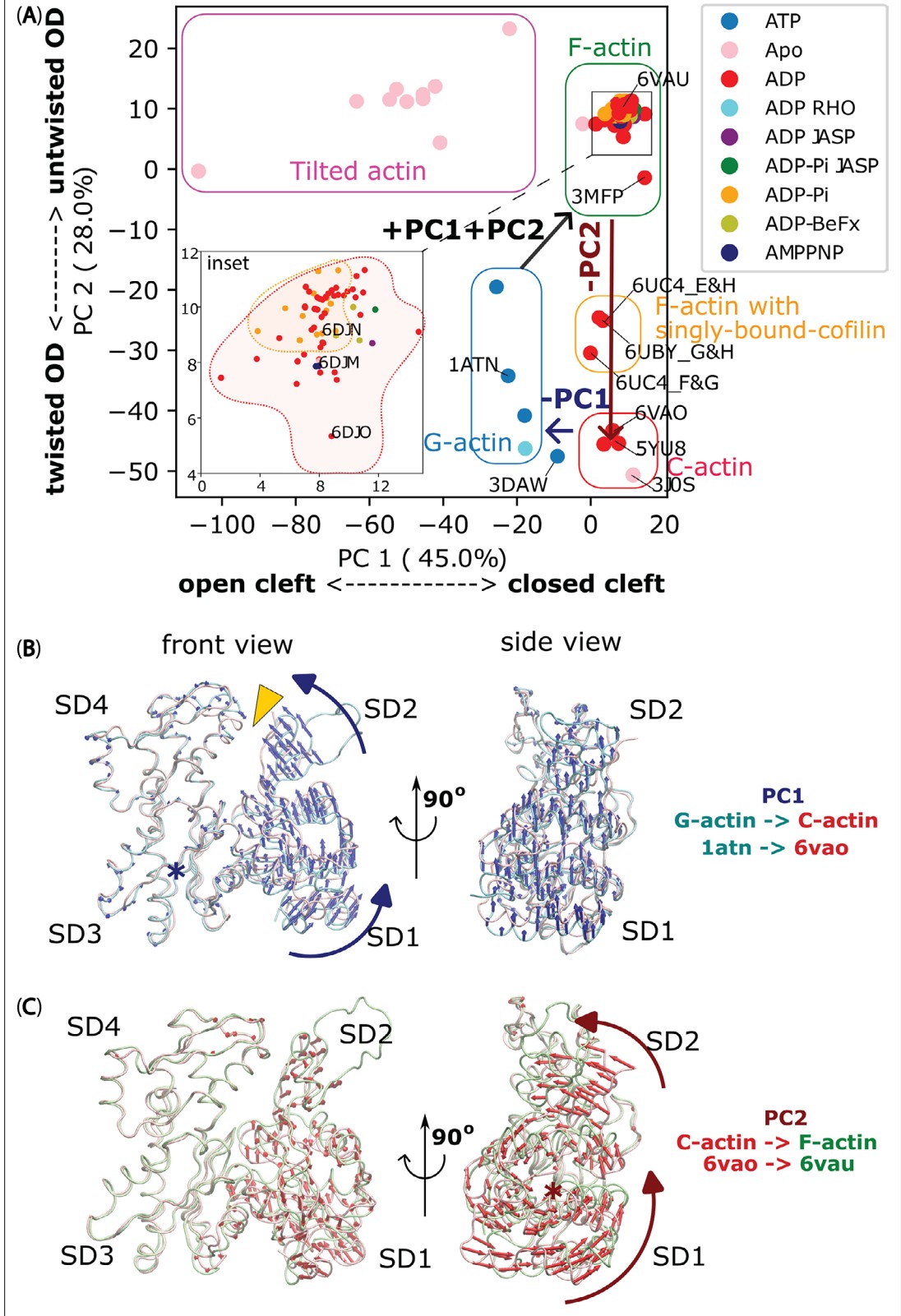

**Figure 1.** The significant movements of the OD during F-actin to C-actin to G-actin transitions, as revealed by PCA. (**A**) PCA of actin chains reported in 46 PDB structures (referred to *Table 1*) reveals collective transition pathway between different actin conformations: G-actin (soluble monomer – blue box), F-actin (filamentous protomer reported with no cofilin bound – green box), C-actin (filamentous protomer reported within cofilactin after removal of cofilin – red box), tilted actin (reported in 3J8J and 3J8K – magenta box) and single-cofilin-bound actin, where one cofilin molecule bound to an

*Figure 1 continued on next page*

*Figure 1 continued*

actin protomer within an actin filament (reported in 6UBY – orange box). Protomers at the most barbed end side of a cofilin cluster in 6UC4 also belong to the latter cluster. Notably, a twinfilin-1 bound structure of soluble actin molecule (3DAW) (*Paavilainen et al., 2008*), previously classified as C-actin (*Tanaka et al., 2018*), was found between the G-actin and C-actin clusters. PDB entries are coloured with nucleotide molecules present, and structures lacking reported nucleotides are labelled 'Apo'. The inset shows a magnified region of the F-actin's PCA cluster (corresponding to the black box in the main graph). (**B**) The PC1, represented by blue arrows, induces a rotational motion of the OD around hinge region, causing the nucleotide binding cleft between SD2 and SD4 (depicted as a yellow arrowhead) to close during the transition from G-actin (cyan chain) to C-actin (pink chain). (**C**) The PC2, indicated by red arrows, results in an untwisting motion of the SD2, leading to the flattening of OD during the transformation from C-actin (pink chain) to F-actin (green chain), but without affecting the closing of nucleotide binding cleft (nearly unchanging PC1). The rotational axes in panels B and C were denoted by blue and red asterisks, respectively.

The online version of this article includes the following source data and figure supplement(s) for figure 1:

**Source data 1.** PC1 and PC2 values obtained from PCA.

**Figure supplement 1.** A closer examination of PC1 and PC2 in the PCA clusters of F-actin structures.

of the OD around the hinge region to seal the nucleotide binding cleft by reducing the SD2-SD4 distance (as indicated by the yellow arrowhead in *Figure 1B* and *Video 1*), depicting the transformation between G-actin and C-actin. Structures with higher PC1 values such as F-actin and C-actin are associated with conformations featuring a closer nucleotide binding cleft. Conversely, structures with lower PC1 values, including tilted-actin and G-actin, are related to conformations featuring a more open nucleotide binding cleft. The second prominent mode (PC2, accounting for 28% of the structural variance) included the twisting of the OD during the transition between F-actin and C-actin (*Figure 1C* and *Video 1*). A higher PC2 value indicates that the OD is more flattened or untwisted, as seen in the tilted-actin and F-actin. On the other hand, a lower PC2 value suggests that the OD is more twisted, as observed in G-actin and C-actin.

Our PCA results indicated that the structure of an actin filament with singly bound cofilin – 6UBY (orange box in *Figure 1A*) was in the transition pathway between the F-actin and C-actin conformations, particularly along PC2. This cluster also includes structures of two protomers positioned at the BE side near a cofilactin cluster including chains E&H combined as E and F&G combined as F in PDB: 6UC4 (*Table 1*). Consistent with previous studies (*Galkin et al., 2011*; *Tanaka et al., 2018*), our analysis suggests that the attachment of a single cofilin to an actin filament induces a slight twisting of the OD relative to the ID (~5°). Furthermore, this effect was cumulative, with more extensive cofilin binding leading to larger OD twisting angles (~10°), thus representing the C-actin state. Throughout this transition, the nucleotide binding cleft remains closed as observed through the minimal variations in PC1 values. This suggests that cofilin prefers to bind to the C-actin structure and the transitional actin states featuring structures resembling C-actin within the actin filaments. However, we also predicted that cofilins do not preferentially bind to the tilted structural state of actin, as tilted actin structures exhibited opened nucleotide-binding clefts, but without the twisting of the OD (*Galkin et al., 2015*) (indicated by lower PC1 and higher PC2 values, as represented by the magenta box in *Figure 1A*). Additionally, these tilted actin structures did not align with the conformational transition pathway between F-actin and C-actin.

Noticeably, PCA analysis revealed higher structural flexibility in F-ADP-actin (red dots), exploring a larger space than F-ADP-Pi-actin structures (orange dots) within the F-actin cluster (inset in *Figure 1A*). This included the F-ADP-actin structure (3MFP; *Fujii et al., 2010*), which notably exhibited a PC2 value significantly approaching that of C-actin structure (*Figure 1A*). However, upon comparison of structures simultaneously resolved in *Chou and Pollard, 2019*, a slightly smaller PC2 value for the F-ADP-actin (6DJO) was observed in contrast to F-AMPPNP-actin (6DJM) and F-ADP.Pi-actin (6DJN; inset in *Figure 1A*). Significantly, the magnified views in the F-actin cluster highlight structural resemblance between F-ADP-actin and F-ADP.Pi-actin in recent studies (*Oosterheert et al., 2022*; *Reynolds et al., 2022*). The F-ADP-actin structure (8D13) and F-ADP.Pi-actin structure (8D14) exhibit remarkably similar PC2 values (*Figure 1—figure supplement 1A*). Furthermore, the PC2 results demonstrate notable similarities between $Mg^{2+}$-F-ADP-actin (8A2T) and $Mg^{2+}$-F-ADP.Pi-actin (8A2S), and $Mg^{2+}$-F-ADP. $BeF_3$-actin (8A2R; *Figure 1—figure supplement 1B*). Altogether, we suggest that the conformation adopted by actin in the cofilactin region, different from actin-nucleotide-solely dependent state, is the most favorable one for binding cofilin.

**Table 1.** List of 46 protein data bank (PDB) structures that were utilized in the PCA, as depicted in *Figure 1*.
Due to the presence of varying structures among different chains in some PDB structures, each chain was considered as an individual structure. Notably, in the case of 6UC4, chains E and H were combined and renamed as chain E (E&H=E), while chains F and G were combined and renamed as chain F (F&G=F). Similarly, in the case of 6UBY, chains G and H were combined and renamed as chain G (G&H=G).

| PDB ID | List of actin chains | Method | Resolution (Å) | Nucleotide | Protein bound | First author | Year |
|---|---|---|---|---|---|---|---|
| 1ATN | A | X-RAY | 2.8 | ATP | DNASE I | KABSCH | 1990 |
| 1J6Z | A | X-RAY | 1.54 | ADP-RHO | | OTTERBEIN | 2001 |
| 2BTF | A | X-RAY | 2.55 | ATP | PROFILIN | SCHUTT | 1993 |
| 2ZWH | A | X-RAY | 3.3 | ADP | | ODA | 2009 |
| 3DAW | A | X-RAY | 2.55 | ATP | TWINFILIN-1 | PAAVILAINEN | 2008 |
| 3HBT | A | X-RAY | 2.7 | ATP | | WANG | 2010 |
| 3J0S | A, B, C, D, E, F, G, H, I, J, K, L | EM | 9 | | COFILIN-2 | GALKIN | 2011 |
| 3J8I | D, E, F, G, H | EM | 4.7 | ADP | | GALKIN | 2015 |
| 3J8J | A, B, C, D, E, F, G, H, I, J, K | EM | 12 | | | GALKIN | 2015 |
| 3J8K | A, B, C, D, E, F, G, H, I, J | EM | 12 | | | GALKIN | 2015 |
| 3MFP | A | EM | 6.6 | ADP | | FUJII | 2010 |
| 4A7N | A, B, C, D, E | EM | 8.9 | ADP | | BEHRMANN | 2012 |
| 5ONV | A, B, C, D, E | EM | 4.1 | ADP | | MERINO | 2018 |
| 5OOC | A, B, C, D, E | EM | 3.6 | ADP JASP | | MERINO | 2018 |
| 5OOD | A, B, C, D, E | EM | 3.7 | ADP-Pi JASP | | MERINO | 2018 |
| 5OOE | A, B, C, D, E | EM | 3.6 | ANP | | MERINO | 2018 |
| 5OOF | A, B, C, D, E | EM | 3.4 | ADP-BeF$_x$ | | MERINO | 2018 |
| 5YU8 | A, B, C, D, E | EM | 3.8 | ADP | COFILIN-2 | TANAKA | 2018 |
| 6BNO | A, B, C, D, E, F, G, H | EM | 5.5 | ADP | | GUREL | 2017 |
| 6BNU | A, B, C, D, E, F, G, H | EM | 7.5 | | | GUREL | 2017 |
| 6DJM | A, B, C, D | EM | 3.1 | AMPPNP | | CHOU | 2019 |
| 6DJN | A, B, C, D | EM | 3.1 | ADP-Pi | | CHOU | 2019 |
| 6DJO | A, B, C, D | EM | 3.6 | ADP | | CHOU | 2019 |
| 6FHL | A, B, C, D, E | EM | 3.3 | ADP-Pi | | MERINO | 2018 |
| 6KLL | A, B, C, D | EM | 3 | ADP | | ODA | 2020 |
| 6KLN | A, B, C, D | EM | 3.4 | ADP | | ODA | 2020 |
| 6UBY | A, B, C, D, E, F, G | EM | 7.5 | ADP | COFILIN-1 | HUEHN | 2020 |
| 6UC0 | A, B, C, D, E, F, G | EM | 7.5 | ADP | COFILIN-1 | HUEHN | 2020 |
| 6UC4 | A, B, C, D, E, F, J, K, L | EM | 9.2 | ADP | COFILIN-1 | HUEHN | 2020 |
| 6VAO | D, A, B, C, E | EM | 3.4 | ADP | COFILIN-1 | HUEHN | 2020 |
| 6VAU | B, A, C, D, E | EM | 3.5 | ADP | | HUEHN | 2020 |
| 7Q8B | A, B, C, D, E | EM | 3.3 | ADP-Pi | | KOTILA | 2022 |
| 7Q8C | A, B, C, D, E | EM | 2.72 | ADP | | KOTILA | 2022 |
| 7Q8S | A, C, D, E, F | EM | 3.4 | ADP | ADF/COFILIN | KOTILA | 2022 |
| 8A2R | C, A, B, D, E | EM | 2.17 | ADP-BeF$_3$ | | OOSTERHEERT | 2022 |

*Table 1 continued on next page*

*Table 1 continued*

| PDB ID | List of actin chains | Method | Resolution (Å) | Nucleotide | Protein bound | First author | Year |
|---|---|---|---|---|---|---|---|
| 8A2S | C, A, B, D, E | EM | 2.22 | ADP-Pi | | OOSTERHEERT | 2022 |
| 8A2T | C, A, B, D, E | EM | 2.24 | ADP | | OOSTERHEERT | 2022 |
| 8A2U | C, A, B, D, E | EM | 2.21 | ADP-BeF$_3$ | | OOSTERHEERT | 2022 |
| 8A2Y | C, A, B, D, E | EM | 2.15 | ADP-Pi | | OOSTERHEERT | 2022 |
| 8A2Z | C, A, B, D, E | EM | 2.15 | ADP | | OOSTERHEERT | 2022 |
| 8D13 | A, B, C | EM | 2.43 | ADP | | REYNOLDS | 2022 |
| 8D14 | A, B, C | EM | 2.51 | ADP-Pi | | REYNOLDS | 2022 |
| 8D15 | A, B, C, D, E, F, G | EM | 3.61 | ADP | | REYNOLDS | 2022 |
| 8D16 | A, B, C, D, E, F, G | EM | 3.71 | ADP-Pi | | REYNOLDS | 2022 |
| 8D17 | A, B, C, D, E, F, G | EM | 3.69 | ADP | | REYNOLDS | 2022 |
| 8D18 | A, B, C, D, E, F, G | EM | 3.66 | ADP | | REYNOLDS | 2022 |

In order to better understand the effect of the collective PC2 movements of individual ODs on the global half helices of filaments, we compared the side views and outer views of F-actin and C-actin protomers in their filamentous structures (*Figure 2A–C*). The twisting movement of OD that we found in *Figure 1C* is best represented again from the side views with the rotation axes (marked as red stars in blue boxes of *Figure 2A–B*). We identified that cofilin clusters rotate the ODs approximately 10°, burying the SD2 including D-loops deeper into the core of the filament. As a result, the twisting angles between two adjacent protomers in the same strand appears to increase from approximately 26° to around 36° if viewed from the outer side (black boxes of *Figure 2A–B*), as reported before (*Galkin et al., 2011*; *Tanaka et al., 2018*). Such change in the twisting angles from the outer views can be represented as the change of the OD lines on 2D surfaces of imaginary cylinders of the filaments, unwrapped along imaginary lines that an AFM tip would scan through (*Figure 2D–E*). Such diagrams depict the varying twisting angles of OD domains within F-actin and C-actin (PC2 mode), leading to the formation of HHPs with different numbers of detected protomer pairs. Hence, collective movements of individual ODs that decrease PC2 values would lead to a globally shorter HHPs and fewer number of protomer pairs per HHP. Thus, this clearly demonstrates that the global shortened HHP is directly related to the OD twists of the local actin protomers (*Figure 2B, C and E*), which represent the C-actin structure.

## The half helices within long actin filaments exhibit heterogeneity in their lengths and the number of protomers per half helical pitch when observed through HS-AFM

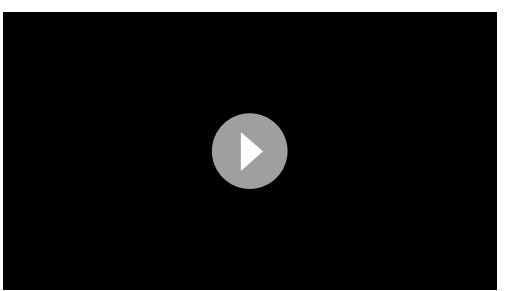

**Video 1.** Possible transition pathways from G-actin to C-actin (constructed by +PC1) and C-actin to F-actin (constructed by +PC2), and vice versa, constructed by -PC1 and -PC2, respectively. The transition from G-actin to F-actin ultimately requires both +PC1+PC2. Related to *Figure 1*.

https://elifesciences.org/articles/95257/figures#video1

In our actual AFM measurements, we determined the half helices, the number of protomer pairs per HHP, and MAD as illustrated in the pseudo-AFM images representing F-actin, C-actin, and cofilactin structures (*Figure 2—figure supplement 1*, *Table 2*, and Methods). To investigate the presence of random variable helical twists in long actin filaments in solution, we employed HS-AFM to examine the dynamic structures and distribution of actin protomers within multiple half helices, with high spatial (1 nm in the xy plane and 0.1 nm in the z-axis) and temporal (2 frames per second) resolution. As depicted in *Figures 3–4*,

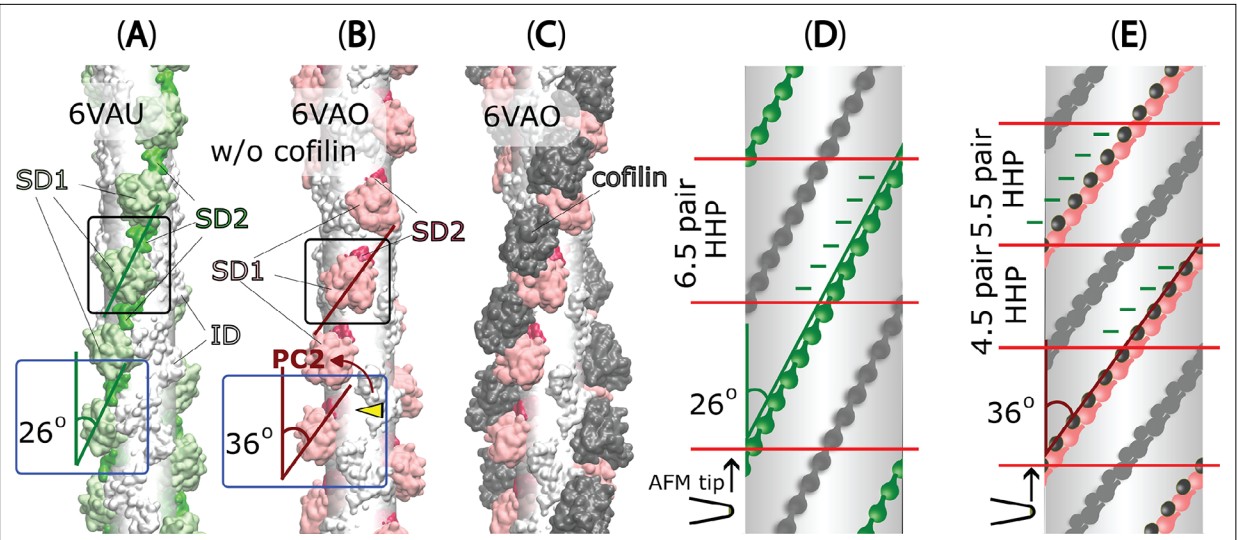

**Figure 2.** The variations in half helices and the number of protomer pairs per HHP in F-actin, C-actin, and cofilactin structures in relationship with PC2 analysis. (**A–C**) Side-views (represented by blue boxes, similar to the right panel of *Figure 1C*) and outer-views (represented by black boxes) of F-actin, C-actin protomers without and with cofilins along long filaments are depicted. The white translucent cylinders represent the central axes of the filaments. In panel B, a yellow arrowhead highlights a portion of SD2 including D-loop that is buried deeper into the C-actin filament as a result of the reduction of PC2 (represented by the red arrow, corresponding to the red arrow in the right panel of *Figure 1C*). (**D–E**) Diagrams of 2D surfaces (unwrapped from imaginary cylinders along the filaments and cut open along imaginary lines scanned by the AFM tip) illustrate the different twisting angles between adjacent protomers in the same strand appears to increase from approximately 26° to around 36° if view from the outer side, resulting in HHPs with varying numbers of detected protomer pairs (marked between red lines). This demonstrates that the global shortened pitch is directly related to the OD twist of the local actin protomers. The green and pink objects represent outer-views of OD domains of the focused chains, along with cofilins shown in black. The opposite chains are depicted in grey.

The online version of this article includes the following figure supplement(s) for figure 2:

**Figure supplement 1.** Pseudo-AFM images featuring long filaments representing F-actin and C-actin structures.

*Table 3*, and *Video 2*, we successfully imaged and quantified the number of actin protomers within many half helices, allowing us to precisely measure the distribution of the number of protomer pairs per HHP (*Figure 3A*). Subsequently, we established a correlation between the HHP and the number of protomer pairs per HHP. We also calculated the HHPs assuming a constant MAD of 5.5 nm and compared them with actual data (*Figure 3B*). We analyzed the linear regression analysis results and noted that the slope derived from our actual AFM measurements was smaller than that computed assuming that the MAD remained unchanged.

**Table 2.** The lists of chains and matrices utilized to build the long filaments.

| PDB ID | Chains | RMSD (Å) | Transformation matrix as Euler angles |
|---|---|---|---|
| 6VAU | BA | 0.000497 | [166.588953–2.1209466e-01–2.62674783e-02] |
| 8A2S | ED | 0.000704 | [-166.93535314–10.63461097 –5.38436149] |
| 8A2T | AB | 0.000704 | [-167.39029871–13.65820025 –6.3200023] |
| 3MFP | AB | 0.000398 | [166.656064 2.13617418e-05 7.68141187e-05] |
| 6UC4 | EF | 0.619515 | [164.58920564–0.207997407 –0.80958005] |
| 6VAO | CA | 0.000699 | [162.515987–4.24094412e-02–1.82361154e-01] |
| 5YU8 | ED | 0.000561 | [-162.100031–4.00777476e-05 4.03030046e-05] |
| 3J0S | KL | 0.001134 | [-162.12014856–0.17228856 –0.18642638] |
| *For cofilactin filament | | | |
| 6VAO* | CAJF | 0.000699 | [162.515987–4.24094412e-02–1.82361154e-01] |

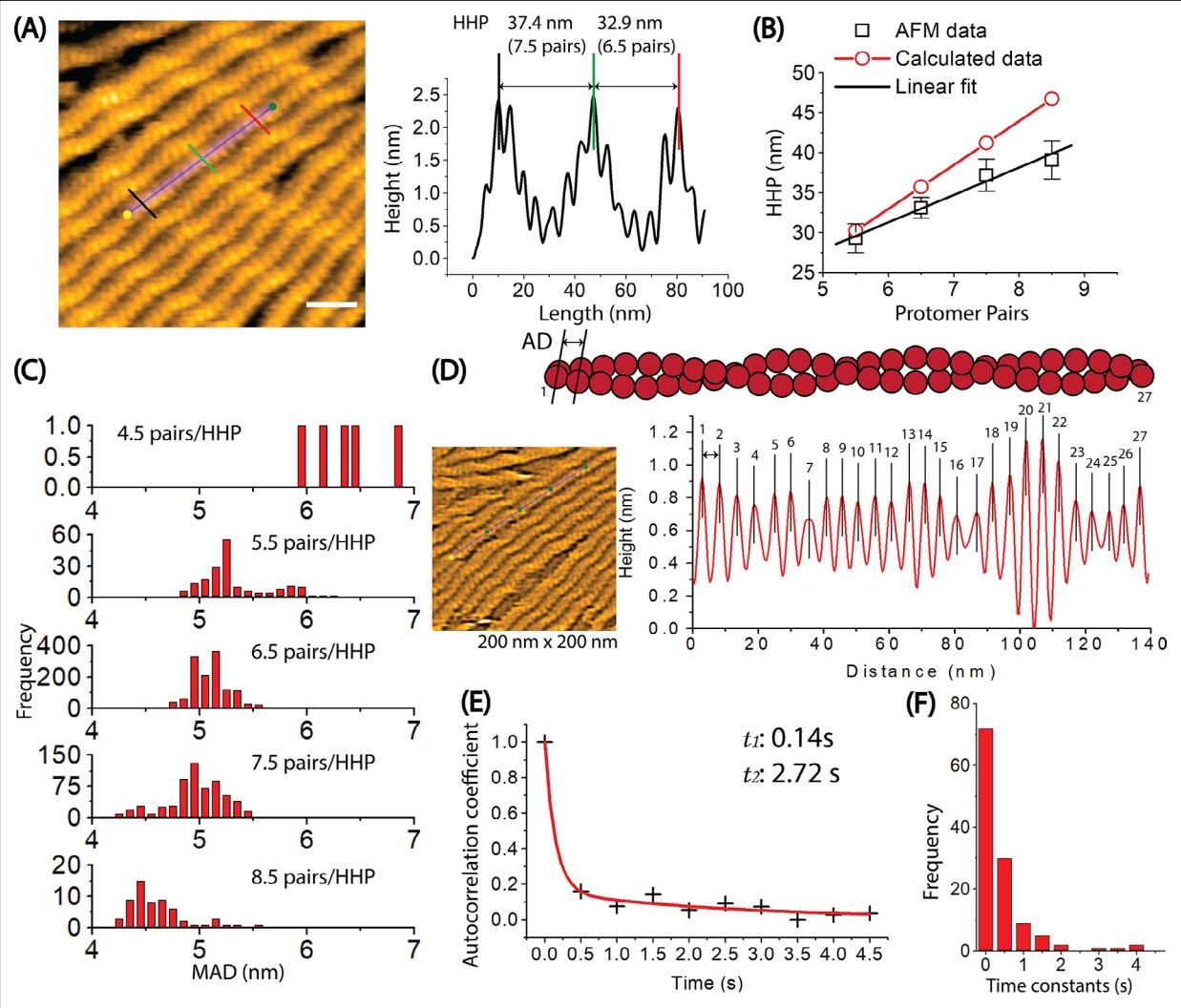

**Figure 3.** Exploring the dynamics of helical structures of long actin filaments using HS-AFM. (**A**) The provided image shows actin filaments with varying numbers of protomer pairs in one half helix. To measure the HHPs and the number of protomer pairs per HHP, a longitudinal section profile of two consecutive half helices was taken as depicted, following the previous method (*Hayakawa et al., 2023*). Actin filaments were gently immobilized on lipid membrane composed of 1,2-dipalmitoyl-sn-glycero-3-phosphocholine (DPPC) and 1,2-dipalmitoyl-3-trimethylammonium-propane (DPTAP) (90/10 wt%) and imaged in F1 buffer containing 0.5 mM ATP. The offset values were set to zero with reference to the height of the lowest position along the longitudinal section lines. The scale bar represents 25 nm. For the detail, see Methods. (**B**) The correlation between the counted protomer pairs from image A and the HHP probed by HS-AFM is shown. Black symbols represent the experimental data, with the linear fit (y=10.71 + 3.43 x, *R*=0.9903). The calculated data (red symbols) also demonstrate a correlation between the number of protomer pairs per HHP and the HHP, which is calculated as 5.5 nm multiplied by the number of counted protomer pairs per HHP. (**C**) The distribution of MADs is shown for half helices composed of different number of counted protomer pairs. The overall MAD (mean ± SD) is 5.1±0.3 nm (n=15021). (**D**) The axial distance (AD) between two adjacent actin protomers along the same long-pitch strand in individual bare actin filaments were directly measured as the peak-to-peak distance in four consecutive HHPs. Each protomer pair within the filament was assigned a number ranging from 1 to 27. The ADs were measured in a time series, with the imaging rate of 0.5 s per frame and a total of 258 frames. These AD values for 27 protomer pairs within the actin filament in a time series of 258 frames were utilized to determine the autocorrelation coefficients. The still AFM image was shown after processing with Laplacian of Gaussian filter. (**E**) An example of the ACF plot displays the autocorrelation coefficients on the y-axis against the time lags on the x-axis. The plotted data was fitted using a second-order exponential decay function with the time constants of ($t_1$, $t_2$), represented by the red curve. (**F**) A histogram was constructed using the time constants obtained from many similar ACF plots and fittings obtained from different actin filaments (n=61), as showcased in (**D, E**). The number of data points was 122. These time constants are associated with the overall trend of time decay in autocorrelation of axial distance between two adjacent protomers in the same protofilament.

The online version of this article includes the following source data and figure supplement(s) for figure 3:

**Source data 1.** Exploring the dynamics of helical structures of long actin filaments using HS-AFM.

**Figure supplement 1.** Exploring the dynamics of axial distance (AD) between two adjacent protomers within the same actin filaments.

**Figure supplement 1—source data 1.** Exploring the dynamics of axial distance (AD) between two adjacent protomers.

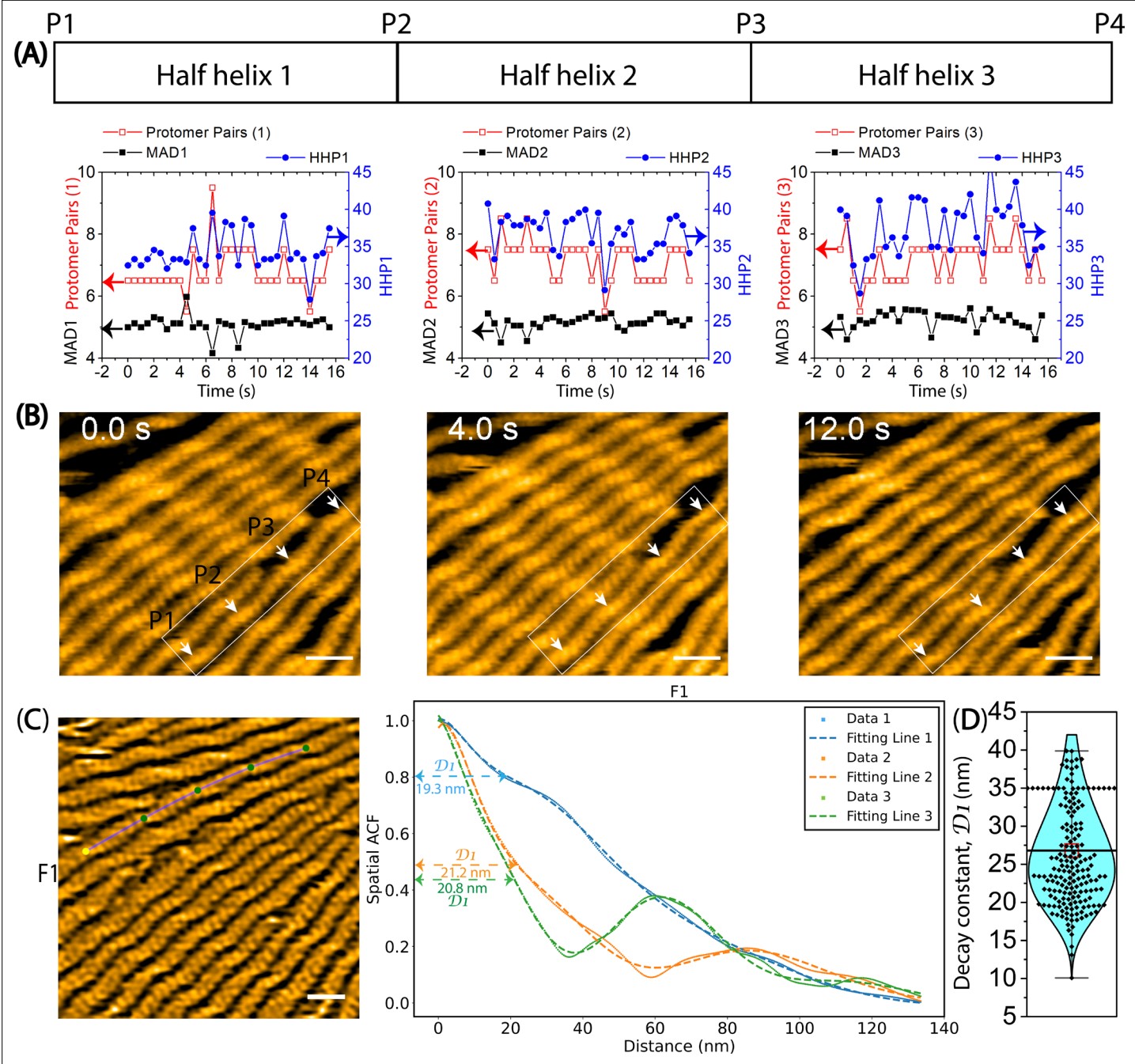

**Figure 4.** The variation of helical twists measured by HHPs, the number of protomer pairs per HHP, and the MADs in long actin filaments through HS-AFM. Actin filaments were gently immobilized on lipid membrane composed of 1,2-dipalmitoyl-sn-glycero-3-phosphocholine (DPPC) and 1,2-dipalmitoyl-3-trimethylammonium-propane (DPTAP) (90/10 wt%) and imaged in F1 buffer containing 0.5 mM ATP. (**A**) The graph demonstrates the changes in HHPs, the number of protomer pairs per HHP, and the MADs over time. The statistical analysis using two-population *t-test* at a significance level of p≤0.05 revealed that the difference between MAD1 (5.1±0.3 nm, n=32) and MAD2 (5.2±0.2 nm, n=32) is not statistically significant (p=0.2365). However, MAD1 and MAD3 (5.2±0.3 nm, n=32) exhibit a slight difference (p=0.02895). (**B**) Representative still images obtained from AFM are presented. Arrows indicate the peaks (**P1, P2, P3, P4**) identified in the raw AFM images. HHP1, HHP2, and HHP3 were measured between P1-P2, P2-P3, and P3-P4, respectively. Scale bars: 25 nm. (**C**) The spatial ACF curves were produced for a canonical actin filament (**F1**) at various time points (i.e. Data 1, 2, 3). The curves were fitted using an exponential decaying sinusoidal function (*Equation 2*), and the resulting decay constant (**D1**) values from individual curves were presented. The still AFM image is shown after processing with Laplacian of Gaussian filter. Bar: 25 nm. (**D**) The *D1* values were determined through a similar fitting of spatial ACF curves, as illustrated in (**C**). This examination was conducted at different time points (with an imaging rate of 0.5 s per frame and a total of 50 frames for each canonical filament (**F1–F4**) using the data, as shown in *Figure 4—figure supplement 2A*). Subsequently, the distribution of *D1* values was analyzed, yielding the mean ± SD value of 26.8±6.8 nm, based on a sample size of 200.

*Figure 4 continued on next page*

*Figure 4 continued*

The online version of this article includes the following source data and figure supplement(s) for figure 4:

**Source data 1.** The variation of helical twists in long actin filaments.

**Figure supplement 1.** A schematic overview of the spatial ACF analysis.

**Figure supplement 1—source data 1.** A schematic overview of the spatial ACF analysis.

**Figure supplement 2.** Analysis of spatial ACF curves.

**Figure supplement 2—source data 1.** Analysis of spatial ACF curves.

To address this discrepancy, we analyzed our measurements, revealing a range of protomer pairs within each HHP, varying from 4.5 to 8.5 with distinct proportions (*Figure 3C*, *Table 3*). The most commonly observed numbers of protomer pairs per HHP were 6.5 and 7.5, accounting for 60.9% (n=1355 HHPs) and 28.0% (n=624 HHPs) of the samples, respectively. Conversely, the least frequently observed numbers of protomer pairs per HHP were 4.5, 5.5 and 8.5, accounting for 0.2% (n=5 HHPs),

**Table 3.** Variation during helical twisting measured by HHPs, the number of protomer pairs per HHP, and the $MAD_f$ observed for protomers representing F-actin structures within long bare actin filaments, as probed by HS-AFM.

The total number of HHPs observed was 2226. The total number of protomer pairs used for estimating $MAD_f$ was calculated by multiplying the number of protomer pairs per HHP by the number of HHPs.

| The number of protomer pairs per HHP | HHPs (nm) | $MAD_f$ (nm) | The number of HHPs (proportion) | The total number of protomer pairs |
|---|---|---|---|---|
| 4.5 | 28.6±1.4 | 6.3±0.3 | 5 (0.2%) | 22.5 |
| 5.5 | 29.3±1.8 | 5.3±0.3 | 182 (8.2%) | 1001 |
| 6.5 | 33.1±1.3 | 5.1±0.2 | 1355 (60.9%) | 8807.5 |
| 7.5 | 37.2±2.0 | 5.0±0.3 | 624 (28.0%) | 4680 |
| 8.5 | 39.1±2.4 | 4.6±0.3 | 60 (2.7%) | 510 |

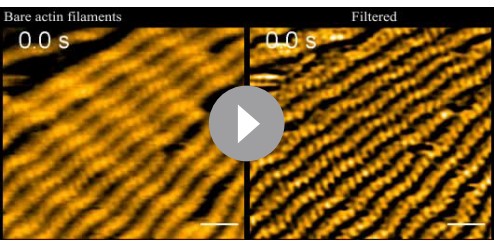

**Video 2.** HS-AFM imaging to investigate the dynamic structures during helical twisting by measuring the variations of HHPs, the number of actin protomers per HHP, and MADs in F-actin structures within long actin filaments. Our results provided strong evidence supporting the variable helical twisting and dynamics in MADs within actin filaments. However, the results contrast the existing rigid and inflexible depiction of helical twists in Cryo-EM and X-ray diffraction structures. Actin filaments were gently immobilized on lipid membrane composed of 1,2-dipalmitoyl-sn-glycero-3-phosphocholine (DPPC) and 1,2-dipalmitoyl-3-trimethylammonium-propane (DPTAP) (90/10 wt%). Bars: 25 nm. Imaging rate: 2 frames per second (fps). Video plays at 5 fps. Related to *Figure 3*.

https://elifesciences.org/articles/95257/figures#video2

8.2% (n=182 HHPs) and 2.7% (n=60 HHPs) of the samples, respectively. Subsequently, we computed the MAD by dividing the length of each measured HHP by the corresponding number of protomer pairs per HHP, thereby establishing a correlation between these variables (*Figure 3C*). The MAD values (mean ± SD) for HHPs containing 6.5 and 7.5 were 5.1±0.2 nm (n=8807.5 pairs) and 5.0±0.3 nm (n=4,680 pairs), respectively. Corresponding values for HHPs with 5.5 and 8.5 protomer pairs were 5.3±0.3 nm (n=1001 pairs) and 4.6±0.3 nm (n=510 pairs) (*Table 3*). Notably, we observed that HHPs with fewer protomer pairs exhibited elongated MADs. Our analysis also indicates an extremely uncommon occurrence, with approximately 0.2% of the actin protomers within the F-actin structure (4.5 protomer pairs per HPP) demonstrating elongated MADs (6.0–6.6 nm).

We next conducted a temporal autocorrelation function (ACF) analysis aimed at estimating the time constants governing the decaying

**Table 4.** Quantifying time constants ($t_1$, $t_2$) obtained from fitting individual ACF curves. The $t_1$ and $t_2$ values are associated with first and second time constants of axial distance (AD) between two adjacent actin protomers within F-ADP-actin and F-ADP.Pi-actin segments. These time constants reflect the time decay in the autocorrelation of AD, recorded experimentally at high temporal resolution using HS-AFM. Related to *Figure 3—figure supplement 1*.

| Experimental conditions | $t_1$ (s) | $t_2$ (s) |
|---|---|---|
| F-ADP-actin (200ms/frame) | 0.4±0.2 (N=82) | 0.5±0.7 (N=80) |
| F-ADP-actin (100ms/frame) | 0.3±0.3 (N=78) | 1.2±1.8 (N=78) |
| F-ADP.Pi-actin (100ms/frame) | 0.2±0.3 (N=152) | 0.5±1.1 (N=141) |

self-correlation in the axial distance (AD) between two adjacent actin protomers along the same long-pitch strand (*Figure 3D–F*, *Figure 3—figure supplement 1*, and *Video 3*). The core objective of this analysis was to determine the temporal extent of AD similarity and elucidate the dynamic behaviors and conformational changes in the distance between adjacent actin protomers over time. The results revealed that self-correlation of the ADs within actin filaments, immobilized on lipid membrane, rapidly decreased within a few hundred milliseconds and then gradually decayed within a few seconds (*Figure 3D–F*, *Figure 3—figure supplement 1*). The $t_1$ and $t_2$ values correspond to the first and second time constant decays of AD between two adjacent actin protomers within F-ADP-actin and F-ADP.Pi-actin segments. At the current imaging rate, these time constants remain very similar between actin filaments bound with ADP or ADP.Pi (*Figure 3—figure supplement 1*, *Table 4*). These results potentially explain the intrinsic dynamics and fluctuations observed in HHPs, the number of protomer pairs per HHP, and the MADs within the actin filaments over time.

To assess the degree of spatial clustering in the structure of actin protomers, we utilized a spatial ACF analysis. This analysis provides an indirect measure of the degree to which height values near each other in space are similar to one another in actin filaments. If protomers can change the shape and/or rotate around the axis parallel to the filament axis, the individual protomers may have slightly different thickness and/or orientations, leading to variations in the heights of protomers and, consequently, variations in the slopes in heights between protomers (depicted in *Figure 4—figure supplement 1*, Methods). As seen in *Figure 4C* and *Figure 4—figure supplement 2A*, the spatial ACF curves were effectively modelled using an exponential decaying sinusoidal function. The decay constant (*D1*) values (mean ± SD) were determined to be 26.8±6.8 nm. This result provided an indirect measure of the degree to which height values near each other in space are similar to one another in canonical actin filaments. Because our AFM images provide detailed surface topography along a longitudinal section profile with outstanding spatial resolution in height (z-axis:~1 Å), we suggest that the similarity in height among protomers in F-actin persists within a region encompassing approximately 5 pairs (~10 actin protomers). However, the limited lateral resolution (x- and y-axis) in our

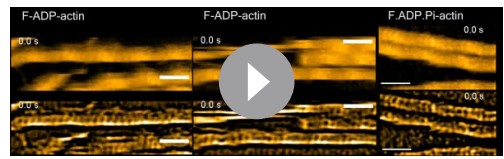

**Video 3.** HS-AFM imaging to investigate the dynamic structures during helical twisting by measuring the variability of MADs in F-ADP-actin and F-ADP.Pi-actin structures within long actin filaments. Our findings provided strong evidence supporting the dynamics in MADs within these actin filaments, though the current temporal resolution could not distinguish a significant difference between F-ADP-actin and F-ADP.Pi-actin, even at high imaging rates. Actin filaments were gently immobilized on lipid membrane composed of 1,2-dipalmitoyl-sn-glycero-3-phosphocholine (DPPC) and 1,2-dipalmitoyl-3-trimethylammonium-propane (DPTAP) (90/10 wt%) and imaged in F1 buffer containing either 1 mM ADP +5 U/ml hexokinase +10 mM glucose or 1 mM ATP +10 mM Pi. Bars: 25 nm. Imaging was conducted at rates of 5 and 10 fps for F-ADP-actin and at 10 fps for F-ADP.Pi-actin. The video is played at 5 fps. Related to *Figure 3—figure supplement 1*.

https://elifesciences.org/articles/95257/figures#video3

AFM images, restricted to 1–3 nm, prevented us from delving further into the structural details of actin protomers within this distance decay.

## The shortened bare HHP adjacent to the cofilin clusters on the PE side includes fewer actin protomers than the normal HHP, resulting in MAD elongation

Using HS-AFM, we aimed to simultaneously capture the helical structures and distribution of actin protomers in the bare F-actin, cofilactin, and hybrid cofilactin/bare actin segments within the same imaging area (see Methods). As shown in *Figure 5—figure supplement 1*, we successfully distinguished cofilactin and bare F-actin structures based on a height difference of approximately 1–1.6 nm and a length difference in the half helices of approximately 25%. A bare F-actin segment consisting of two consecutive half helices (HHP1, HHP2) exhibited the expected normal lengths of 36 nm (6.5 pairs) and 36.5 nm (6.5 pairs; *Figure 5—figure supplement 1A*). In contrast, a cofilactin segment with two consecutive half helices (HHP3, HHP4) displayed shorter lengths of 28.6 nm (4.5 pairs) and 29.8 nm (4.5 pairs; *Figure 5—figure supplement 1B*). Next, we presented a typical segment of the hybrid bare actin/cofilactin structure containing two consecutive half helices (HHP5, HHP6), both showing comparably shortened lengths of 27.9 nm (4.5 pairs) and 30.5 nm (4.5 pairs; *Figure 5—figure supplement 1C*). Based on our previous study (*Ngo et al., 2015*), we judged that the bare half helix with the shorter HHP adjacent to the cofilin cluster is on the PE side and that the normal HHP is located on the BE side. These typical data suggest that, similar to cofilactin segments, shortened bare half helices adjacent to cofilin clusters on the PE side include fewer actin protomers than normal half helices. We next analyzed the MAD variation in this hybrid filament over time and made interesting observations (*Figure 5—figure supplement 2*). We discovered that MAD5 in bare actin HHP5 was elongated similarly to that in cofilactin HHP6 and HHP7. Notably, MAD5 (5.7±0.5 nm, n=13) was extended and closely resembled MAD6 (5.9±0.3 nm) and MAD7 (6.1±0.3 nm). This finding suggests that the MAD elongation in the cofilactin region was transmitted beyond the cofilactin/bare actin boundary to extend MAD5 in the shortened bare half helix.

We then conducted similar analyses on the hybrid cofilactin/bare actin filaments (*Figure 5*, *Figure 6*, *Video 4*). First, we observed the minimal impact of the lipid membrane on the half helices (HHP1, HHP2), number of protomer pairs per HHP, and MADs (MAD1, MAD2) in normal actin filaments (*Figure 5A*). We examined the bare half helices (HHP3–HHP6) and the number of protomer pairs within each HHP on either side of a small cofilin cluster and found that the shortened bare half helix on the PE side includes fewer actin protomers than the normal half helix (*Figure 5B–C*). For example, the cofilin cluster (estimated as 2–4 molecules) shortened one adjacent bare half helix (HHP6) on the PE side (25.5 nm, 4.5 pairs), while the half helix (HHP5) on the BE side was similar to the normal half helix (37.9 nm, 7.5 pairs; *Figure 5C*). In this case, the MAD6 calculated within the shortened bare half helix (HHP6) was 5.7 nm and was larger than the MAD5 value of 5.1 nm calculated within the normal half helix (HHP5; *Figure 5C*). Despite a complete decoration requiring 9–11 cofilin molecules in a half helix, the observation that 2–4 molecules can initiate the supertwisting of a half helix, achieved by reducing the number of protomer pairs per HHP, strongly implies the presence of allosteric transitioning in the supertwisting structures from the bound to the unbound region on the PE side. We anticipate that the unbound region may contain approximately 1–2 pairs (2–4 protomers) resembling C-actin structures. When analyzing the fluctuation in the MADs over time in this hybrid filament, we observed that MAD6 (5.4±0.3 nm, n=12) was significantly greater than MAD5 (5.0±0.2 nm, n=12) (*Figure 6A and B*). Our analysis suggests that the MADs are longer in the shorter bare half helices than in the normal half helices (*Table 5*). Furthermore, we analyzed the fluctuation of the MAD in the F-ADP-actin segments on both sides of large cofilin clusters over time (*Figure 6—figure supplement 1*, *Video 5*). Our analysis suggests that the MADs fluctuate over time and skew towards longer ranges in a supertwisted bare half helix on the PE side compared to an untwisted or normal half helix on the BE side.

We also analyzed a spatial ACF to indirectly measure of the degree to which height values near each other in space are similar to one another, specifically focusing on regions from bound to unbound cofilin. In contrast to the normal actin filament (*Figure 4C*, *Figure 4—figure supplement 2A*), the presence of small cofilin clusters led to a more pronounced linear decline in the spatial ACF curves for each half helix (*Figure 6C*, *Figure 4—figure supplement 2B*). The distribution of the decay

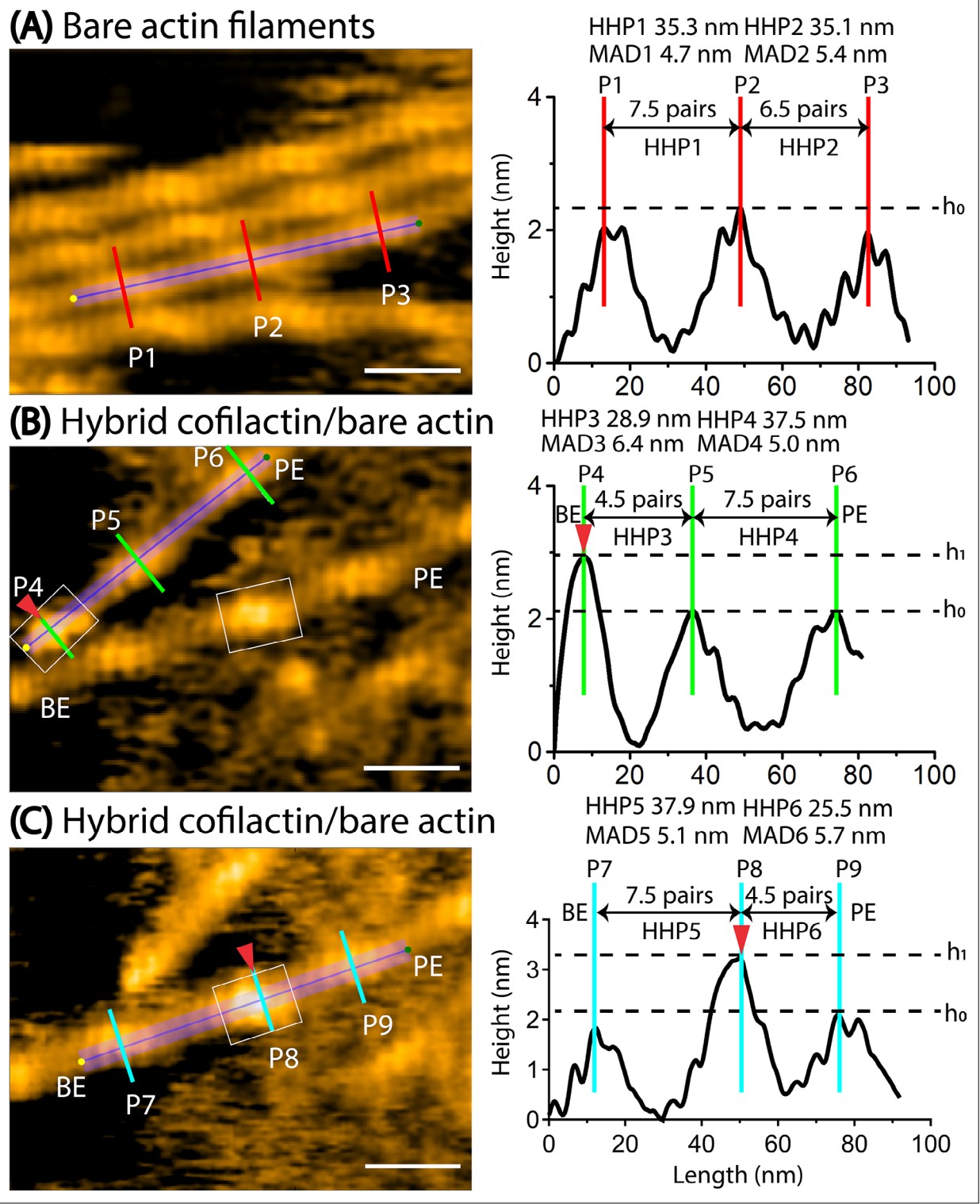

**Figure 5.** Asymmetric helical twisting on both sides of a small cofilin cluster. Red arrowheads indicate small area of cofilin clusters (in boxes). Scale bars: 25 nm. Based on our previous study (*Ngo et al., 2015*), we judged that the bare half helix with the shorter HHP adjacent to the cofilin cluster is on the PE side and that the normal HHP is located on the BE side. (**A**) A still AFM image shows normal actin filaments, along with a longitudinal section profile representing two consecutive half helices 1 and 2 (denoted HHP1, HHP2). (**B**) A still AFM image displays a hybrid segment consisting of cofilactin and bare actin areas, with a longitudinal section profile representing two consecutive half helices. A short filament was observed where a small cofilin cluster cut filament at cofilactin/bare actin boundary on the BE side, followed by two consecutive bare half helices 3 and 4 (denoted HHP3, HHP4) on the PE side. The shortened bare half helix 3 and normal bare half helix 4 are presented. The hybrid cofilactin/bare actin filaments, priorly made in solution, were

*Figure 5 continued on next page*

*Figure 5 continued*

strongly immobilized on lipid membrane composed of 1,2-dipalmitoyl-sn-glycero-3-phosphocholine (DPPC) and 1,2-dipalmitoyl-3-trimethylammonium-propane (DPTAP) (50/50 wt%) and imaged in F1 buffer containing 0.5 mM ATP (see Methods). (**C**) A still AFM image depicts a hybrid segment comprising cofilactin and bare actin areas, along with a longitudinal section profile representing two consecutive half helices 5 and 6 (denoted HHP5, HHP6). In this case, a short filament contained a small cofilin cluster (estimated as 2–4 molecules) in the middle, configured by a normal bare half helix 5 on the BE side and a shortened bare half helix 6 on the PE side. We estimated the size of a cofilin cluster by measuring its height and width within the white box around P8, distinguishing it with the control bare section lacking cofilin bindings (**P7, P9**). Despite the invisibility of cofilin molecules below, we made the assumption that cofilin bound uniformly to two strands at a molar ratio of 1:1, enabling us to account for their presence.

The online version of this article includes the following source data and figure supplement(s) for figure 5:

**Source data 1.** Asymmetric helical twisting on both sides of a small cofilin cluster.

**Figure supplement 1.** The propagation of the supertwisted half helices induced by cofilin clusters results in the shortening of a bare half helix beyond the cofilactin/bare actin boundary.

**Figure supplement 1—source data 1.** The propagation of the supertwisted half helices induced by cofilin clusters.

**Figure supplement 2.** The analysis of the variation in helical twisting by measuring HHPs, the number of protomer pairs per HHP, and MADs in a hybrid bare actin/cofilactin segment using HS-AFM.

**Figure supplement 2—source data 1.** The variation in helical twisting in a hybrid bare actin/cofilactin segment.

constant (*D1*) values was analyzed, yielding the mean ± SD values for each half helix on the PE and BE side of 12.7±7.0 nm and 12.6±6.6 nm, respectively. We propose that the similarity in height among protomers in the bound cofilin region equally extends to the unbound region on the PE and BE sides, encompassing approximately 1–2 pairs (~2–4 actin protomers). Nevertheless, the restricted lateral resolution in our AFM images hindered a detailed exploration of the structural intricacies of unbound actin protomers on the PE or BE side. We propose the future utilization of Cryo-EM to examine the structural details of bare protomers adjacent to a cofilin cluster within this hybrid cofilactin/bare actin filament, which is attached similarly to a supported lipid membrane.

The transmission of the supertwisted helical structure within the cofilin cluster to the adjacent bare half helices is essential in facilitating the preferential cooperative binding of cofilin.

The cooperative binding of cofilin to actin filaments has been previously reported (*Bobkov et al., 2006*; *McGough et al., 1997*; *Ngo et al., 2016*; *Ngo et al., 2015*). However, the role of cooperative conformational changes in actin filaments in driving the preferential binding of cofilin toward the PE side remains unclear, as indicated in several notable studies (*Bibeau et al., 2021*; *Bobkov et al., 2006*; *Galkin et al., 2011*; *Hayakawa et al., 2014*; *Huehn et al., 2018*; *Huehn et al., 2020*; *McCullough et al., 2008*; *Narita, 2020*; *Ngo et al., 2015*; *Suarez et al., 2011*; *Tanaka et al., 2018*; *Wioland et al., 2017*).

We observed the cooperative binding of cofilin to actin filaments when they were loosely absorbed onto a lipid membrane in an F-buffer with cofilin, ATP, and inorganic phosphate (Pi) (*Figure 7A*, *Video 6*). We also examined the peak heights and lengths of the half helices in the cofilactin and bare actin regions during the unidirectional growth of the cofilin clusters (*Figure 7A–C*). Consistent with our previous findings (*Ngo et al., 2015*), the cofilin cluster binding led to an approximately 1–2 nm increase in the peak height and an approximately 25% decrease in the HHPs in the cofilactin region. Then, we analyzed the relationship between the time needed to shorten the bare half helix beyond the cofilactin/bare actin boundary and to form a newly matured cofilin cluster within the shortened bare half helix. Finally, we determined the lag time for the growth of the cofilin cluster within the individual shortened bare half helices adjacent to the cofilin cluster.

The analysis suggested that even with 10 mM Pi, the cofilin cluster could supertwist the half helices in the actin filaments. Notably, the supertwisted half helices within the cofilactin region propagated beyond the cofilactin/bare actin boundary, shortening the adjacent bare half helix before the saturated cofilin cluster grew (*Figure 7A*, *Figure 7B*, *Table 6*). For example, when the HHP between P2 and P3 was shortened at 2 s, P2 was bound to a cofilin cluster, but P3 remained unbound. A lag time of approximately *t*=3 s was needed for the saturated cofilin cluster to be recruited and grew in P3 at 5 s. This pattern was also observed when the HHPs between P3-P4, P4-P5, and P5-P6 shortened before the cofilin clusters expanded. These findings suggest that the shortening of the bare half helix beyond the cofilactin/bare actin boundary drives the actin structure-dependent preferential cooperative binding of cofilin clusters. The cooperative conformational changes during helical twisting may

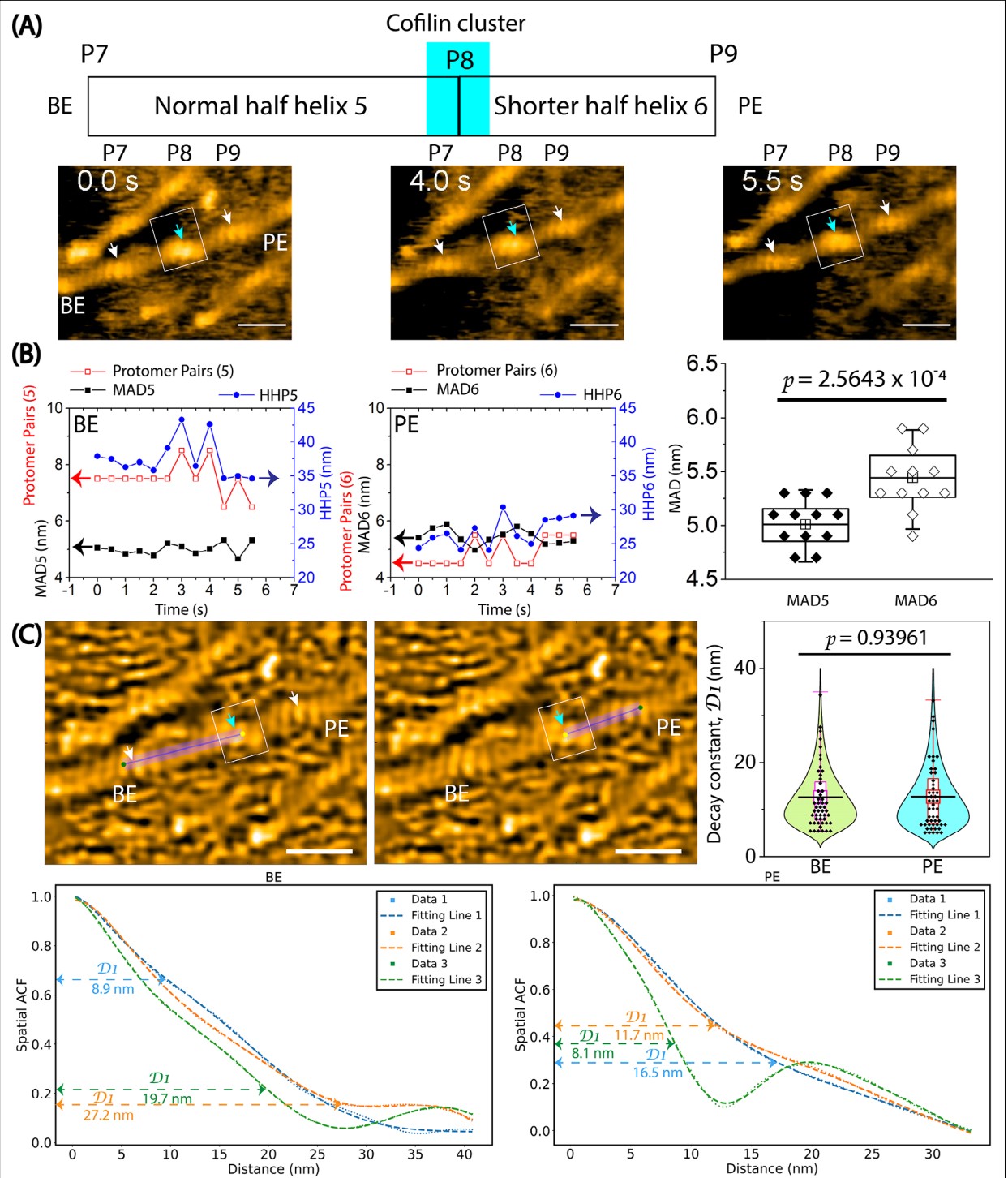

**Figure 6.** The variation of half helices, the number of protomer pairs per HHP, and the MADs in a hybrid cofilactin/bare actin filament probed by HS-AFM. Based on our previous study (***Ngo et al., 2015***), we judged that the bare half helix with the shorter HHP adjacent to the cofilin cluster is on the PE side and that the normal HHP is located on the BE side. (**A**) Representative still images obtained from AFM in ***Figure 5C*** are displayed. Arrows indicate the peaks (**P7, P8, P9**) identified in the raw AFM images. A normal bare half helix 5 (denoted HHP5) and a shortened bare half helix 6 (denoted HHP6) were measured between P7-P8 and P8-P9, respectively. Cyan and white arrows indicate the peak positions within cofilin clusters and bare actin peaks, respectively. Boxes denote the cofilin clusters. Scale bars are 25 nm. (**B**) The graph depicts the analysis of half helices 5 and 6, the number of protomer pairs in each half helix (Protomer Pairs (5) and Protomer Pairs (6)), and the MAD5 and MAD6 over time. Two-populations *t-test* was used to validate the statistical difference between MAD5 (5.0±0.2 nm) and MAD6 (5.4±0.3 nm). The sample size was 12. (**C**) The spatial ACF curves were generated for a normal half helix 5 and a shorter half helix 6 near a cofilin cluster on either the PE or BE side in a hybrid cofilactin/bare actin filament at various

*Figure 6 continued on next page*

*Figure 6 continued*

time points (i.e. Data 1, 2, 3). The curves were fitted using an exponential decaying sinusoidal function (**Equation 2**), and the resulting decay constant (**D1**) values from individual curves were presented. This examination was conducted at different time points (with an imaging rate of 0.5 s per frame and a total of 50 frames for each half helix on the PE or BE side using the data, as shown in **Figure 4—figure supplement 2B**). The *D1* values were determined through a similar fitting of spatial ACF curves. Subsequently, the distribution of *D1* values was analyzed, yielding the mean ± SD values for each half helix on the PE and BE side of 12.7±7.0 nm and 12.6±6.6 nm, respectively. The sample size of each case was 50. The statistical analysis using two-population *t-test* at a significance level of p≤0.05 revealed that the difference between two mean values was not significant. The still AFM images are shown after processing with Laplacian of Gaussian filter. Scale bars: 25 nm.

The online version of this article includes the following source data and figure supplement(s) for figure 6:

**Source data 1.** The analysis of a hybrid cofilactin/bare actin filament.

**Figure supplement 1.** The fluctuation of the mean axial distance (MAD) between two adjacent actin protomers along the same long-pitch strand of F-ADP-actin segments as a function of time.

**Figure supplement 1—source data 1.** The fluctuation of the mean axial distance (MAD) of F-ADP-actin segments.

provide a mechanism for understanding the preferential binding of cofilin clusters toward the PE side, as reported in previous studies (*Bibeau et al., 2021*; *Ngo et al., 2015*).

As negative control experiments, we immobilized actin filaments onto lipid membranes (DPPC/DPTAP = 75/25 wt%) to physically hinder the changes with helical twisting. We then added a high concentration of cofilin (2000 nM) and observed its cooperative binding with actin filaments in the presence of ATP (*Figure 8A*, *Video 7*). However, the binding and expansion of cofilin clusters on these actin filaments were severely inhibited during long incubation times. Instead of cooperatively forming large clusters, cofilin formed only a few small clusters that transiently associated and disso-ciated, even at cofilin concentrations much higher than $k_d$ ($k_d$ = ~5–10 nM; *Andrianantoandro and Pollard, 2006*). The histograms of the peak heights and HHPs were analyzed and showed single peaks at 8.2±0.8 nm (n=1930) and 34.3±2.6 nm (n=1556), respectively, which were similar to those of normal actin filaments (*Table 7*). The decay constant (*D1*) values were determined to be 26.8±6.3 nm (*Figure 8—figure supplement 1*), suggesting that the resemblance in height among protomers in sparsely decorated cofilin-actin filaments extends within a region covering approximately 5 pairs (~10 actin protomers). These values were found to be comparable to those (26.8±6.8 nm) observed in normal actin filaments (*Figure 4C*). Together, our result strongly suggests that the cooperative binding of cofilin is greatly influenced by cooperative conformational changes during helical twisting.

## Examining the inhibitory effect of Pi on the binding and expansion of cofilin clusters

Next, we examined the nucleotide effect on the cooperative binding of cofilin to actin filaments (*Figure 8B–D*, *Videos 8–10*). We analyzed the time-dependent changes in the peak heights and HHPs of representative actin filaments under each condition to track the growth of the cofilin clusters. We made actin filaments incubated with different nucleotides (ADP and ATP +Pi), leading to the formation of F-ADP-actin and F-ADP.Pi-actin, and subsequently mixed with cofilin. We consistently observed the preferential binding of cofilin to shortened bare HHPs toward the PE side. However, we noted that in rare instances, cofilin clusters also grew on both sides in the regular bare half helices when only ADP was present. The histograms of the peak heights and HHPs followed Gaussian distributions corresponding to the cofilactin and normal bare actin regions (*Figure 8B–C*, *Table 7*). The growth rates of the cofilin clusters within the shortened bare half helices adjacent to the cofilin clusters in both F-ADP.Pi-actin and F-ADP-actin were not significantly different, as shown in *Figure 9A–B*. Currently, we have no experimental data to discuss about whether and when bound Pi is released during this process. However,

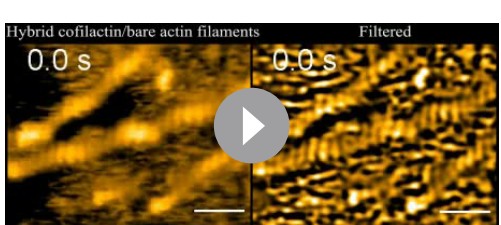

**Video 4.** HS-AFM imaging to investigate the dynamic structures during helical twisting by measuring the variations of HHPs, the number of actin protomers per HHP, and MADs in the normal and shorter half helices neighboring a cofilin cluster on the BE and PE sides. Bars: 25 nm. Imaging rate: 2 fps. Video plays at 5 fps. Related to *Figures 5–6*.

https://elifesciences.org/articles/95257/figures#video4

**Table 5.** Variation during helical twisting measured by HHPs, the number of protomer pairs per HHP, and the $MAD_c$ observed for protomers resembling C-actin structures within shortened bare half helices adjacent to the cofilactin/bare actin boundary, as probed by HS-AFM.

The total number of HHPs observed was 32. The total number of protomer pairs used for estimating $MAD_c$ was calculated by multiplying the number of protomer pairs by the number of HHPs.

| The number of protomer pairs per HHP | HHP (nm) | $MAD_c$ (nm) | The number of HHPs (proportion) | The total number of protomer pairs |
|---|---|---|---|---|
| 3.5 | 22.2±0.2 | 6.3±0.1 | 2 (5.9%) | 7 |
| 4.5 | 26.1±1.4 | 5.8±0.3 | 21 (61.8%) | 94.5 |
| 5.5 | 28.5±0.9 | 5.2±0.2 | 9 (32.4%) | 49.5 |

three major processes could be considered once a cofilin cluster with a critical size has formed, the allosteric transmission of the supertwisted structure of the half helices within the cofilactin region to an adjacent bare half helix on the PE side results in C-actin-like protomers. First, Pi is released accompanying (preceding) supertwisting structural change in the bare region near cofilin clusters on the PE side. Because most structural studies suggest that cofilin binding promotes Pi release, leading to the formation of C-actin with ADP binding (*Figure 1A*), we anticipate that protomers within the shortened bare half helix adjacent to cofilin clusters in F-ADP.Pi-actin rapidly converts to F-ADP-actin and C-actin-like structures, thereby facilitating the expansion of cofilin clusters toward the states of actin bound to ADP. Second, Pi remains bound to protomers within the supertwisted bare section. The successive interactions with cofilin molecules promote Pi release. Third, similar to the process described in the second one, but in this case, subsequent interactions with cofilin molecules do not induce the release of Pi. Consequently, numerous actin protomers within the newly formed cofilactin region remain bound to ADP.Pi. When we observed neighboring bare half helices on both sides of a cofilin cluster in F-ADP-actin and F-ADP-Pi-actin within one imaging field, the HHP was always shorter on one side, while the HHP was essentially normal on the other side in both F-ADP-actin and F-ADP. Pi-actin. These results suggest that bound Pi has a minor effect, hindering the propagation of the supertwisted helical structure within cofilin clusters to the neighboring bare half helices on the PE side.

To examine how bound Pi inhibits cofilin binding (*Blanchoin and Pollard, 1999*; *Carlier et al., 1997*; *Carlier and Pantaloni, 1988*), we assessed the torsional flexibility by measuring the HHPs in control actin filaments incubated with ADP and ATP +Pi, or ATP +phalloidin (*Figure 9C–D*). In all cases, the histograms followed typical HHP distributions. The mean HHPs were nearly identical. However, the variances of the HHPs in F-ADP.Pi-actin and phalloidin-stabilized F-actin were significantly smaller than that in F-ADP-actin. Consequently, the proportion of naturally supertwisted half helices with HHPs shorter than 30 nm was 5.8% for F-ADP-actin but only 1.1% and 0.2% for F-ADP.Pi-actin and phalloidin-stabilized F-actin, respectively. Our finding provides the structural support for earlier research that the nucleotide state of actin could influence F-actin's flexibility and its ability to undergo rearrangements that mediate ABP binding (*Blanchoin and Pollard, 1999*; *Okura et al., 2023*; *Suarez et al., 2011*; *Zimmermann et al., 2015*). This is in line

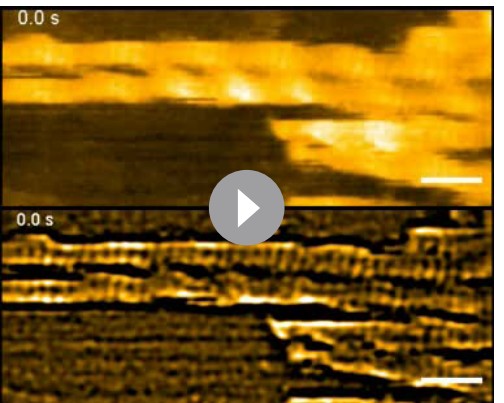

**Video 5.** HS-AFM imaging to investigate the dynamic structures during helical twisting by measuring the variations of MADs in the barely supertwisted and untwisted half helices neighboring a cofilin cluster on the PE and BE sides, respectively, as well as in the normally twisted half helices in the same images. Actin filaments were imaged in an F1 buffer containing 300 nM cofilin and 1 mM ADP, 5 U/ml hexokinase and 10 mM glucose. There was minimal nonspecific binding of cofilin to this lipid membrane surface. Bars: 25 nm. Imaging rate: 2 fps. Video plays at 5 fps. Related to *Figure 6—figure supplement 1*.

https://elifesciences.org/articles/95257/figures#video5

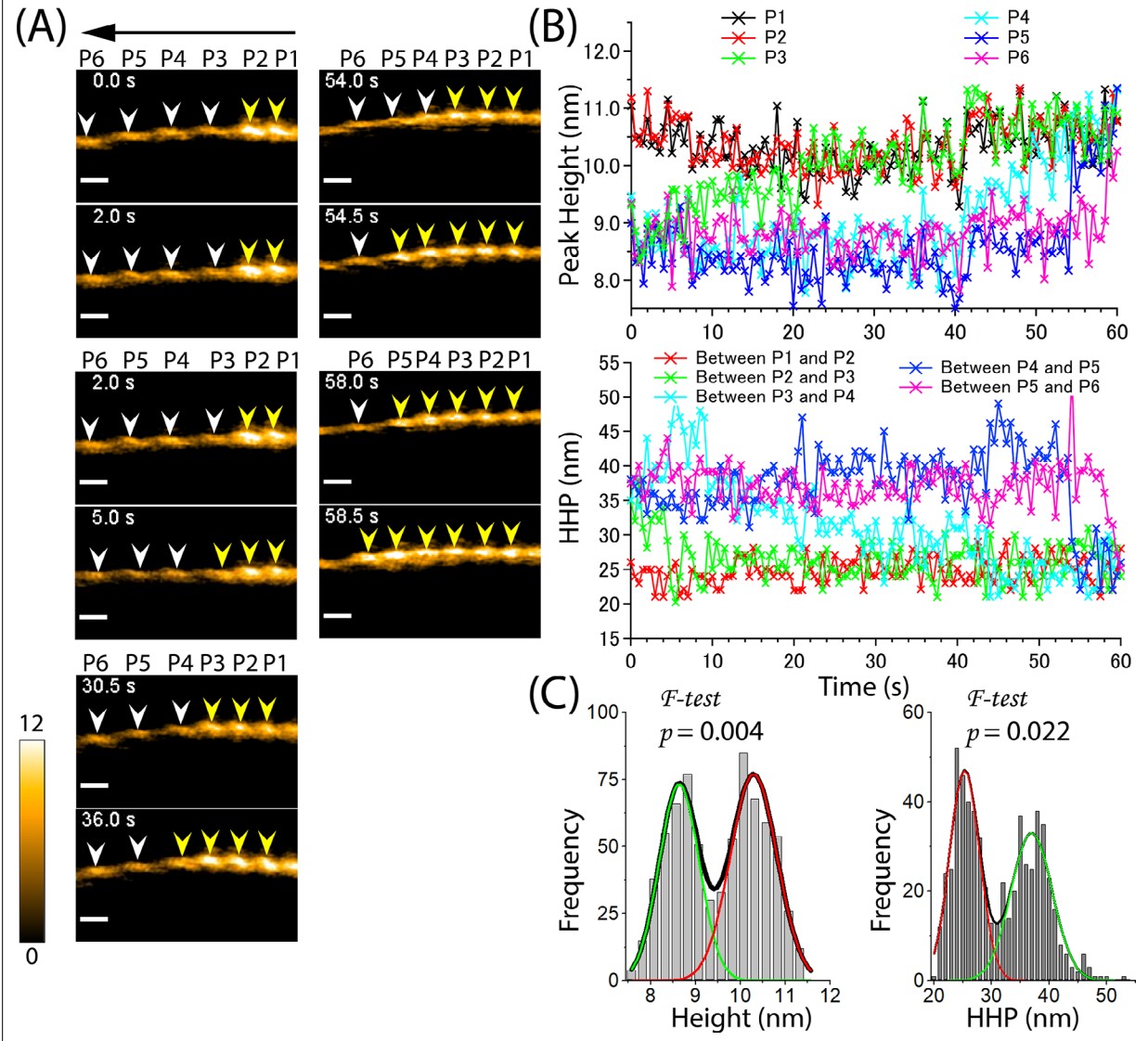

**Figure 7.** Propagation of supertwisted half helices within the cofilactin region beyond the boundary of cofilactin and bare actin, leading to the pre-shortening of a bare half helix and subsequent recruitment of cofilin cluster growth. (**A**) Still AFM images show the time-dependent growth of cofilin clusters beyond the boundary. The arrow indicates the direction of cofilin cluster growth. The peaks of cofilactin and bare actin segments are marked by yellow and white arrowheads, respectively. Peak heights (**P1–P6**) were determined in five consecutive half helices. Actin filaments were loosely immobilized on a lipid membrane composed of 1,2-dipalmitoyl-sn-glycero-3-phosphocholine (DPPC) and 1,2-dipalmitoyl-3-trimethylammonium-propane (DPTAP) (90/10 wt%) and imaged in an F1 buffer containing 300 nM cofilin, 1 mM ATP and 10 mM Pi. Scale bars: 25 nm. (**B**) Examination of changes in peak heights and HHPs in the cofilactin and bare actin regions over time. Peak heights and HHPs were measured as described previously (*Ngo et al., 2015*). (**C**) Histograms of peak heights (8.6±0.4 nm, 10.3±0.5 nm, n=726) and HHPs (25.3±2.6 nm, 36.9±3.6 nm, n=605) were obtained from the data in (**B**). The significant difference between the peaks in double Gaussian fitting was statistically examined using an *F-test*. The green and red curves denote the fractions of F-actin and cofilactin structures, respectively.

The online version of this article includes the following source data for figure 7:

**Source data 1.** Propagation of supertwisted half helices within the cofilactin region beyond the boundary of cofilactin and bare actin.

with measurements of micrometer-scale persistence length, indicating that F-ADP.Pi-actin is more rigid than F-ADP-actin (*Isambert et al., 1995*), and with the mechanical control of twisting and bending of actin's nucleotide-bound states (*De La Cruz et al., 2010*). Assuming that initial cofilin binding occurs at sites with naturally supertwisted helical structures (*Galkin et al., 2001*), we suggest that bound Pi in F-ADP.Pi-actin mainly inhibits initial cofilin binding by reducing the fraction of supertwisted helical structures in comparison to F-ADP-actin.

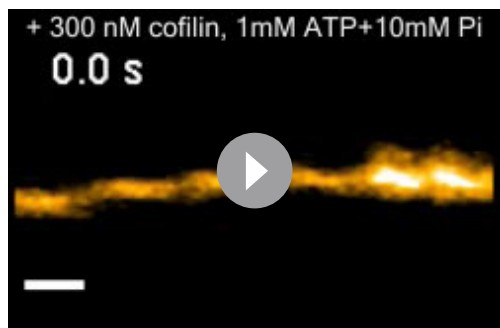

**Video 6.** HS-AFM imaging to demonstrate the preferentially cooperative binding of cofilin toward a pre-shortened bare half helix resembling C-actin structures along actin filaments. Even with 10 mM Pi, the cofilin cluster could supertwist the half helices in the actin filaments once a critical cluster was established. Bars: 25 nm. Imaging rate: 2 fps. Video plays at 5 fps. Related to *Figure 7*.

https://elifesciences.org/articles/95257/figures#video6

## Discussion

We investigated the dynamic conformational changes in half helices of F-actin and C-actin-like structure segments by observing the fluctuations in the number of actin protomers per HHP and MAD. The results provide robust support for a prior study (*Egelman et al., 1982*), which reported variable helical twisting within actin filaments. Nonetheless, the Cryo-EM and X-ray diffraction findings depict the rigid and inflexible helical twists of actin filaments (HHP = 36 nm). Certainly, HS-AFM captures the dynamic structural changes of proteins during functional processes in solution, offering a trade-off with lower spatial resolution but superior temporal resolution compared to Cryo-EM (*Ando, 2012*; *Ando et al., 2008*; *Ando et al., 2024*). We leverage the advantage of this technique to capture the torsional flexibility during helical twisting of actin filaments and elucidate their adaptable structure as a variable helical twist, which plays crucial roles in the binding and cluster expansion of cofilin. While it was previously believed that the interaction between the upper and lower clefts establishes the structural basis for how nucleotide-dependent conformation changes in F-actin structure modulate the binding affinities for ABPs including cofilin (*Dominguez and Holmes, 2011*), recent structural analyses have revealed a remarkable similarity in the average structure of actin protomers bound with ADP.BeF$_3$, ADP.Pi, and ADP in the presence of Mg$^{2+}$ (*Oosterheert et al., 2022*; *Reynolds et al., 2022*; *Figure 1—figure supplement 1*), significantly challenging the hypothesis regarding the modulation of ABP binding affinities through nucleotide-dependent conformational changes. So, why is the flexibility in torsion during helical twisting crucial for the preferential cooperative binding of cofilin to actin filaments toward the PE side? Unfortunately, we currently lack the atomic structures of actin protomers adjacent to cofilin clusters on the PE side to fully address this question.

Nonetheless, our PCA results reveal that key longitudinal interactions between the D-loop of SD2 positioned in the pocket of the next protomer's SD1 contribute to keeping the nucleotide binding

**Table 6.** The correlation between the time it takes to shorten a bare HHP beyond the cofilactin/bare actin boundary and to form a newly matured cofilin cluster in a pre-shortened half helix.

The lag time (*t*) represents the duration required for a newly saturated cofilin cluster to bind and decorate onto a bare half helix adjacent to the preformed cluster after this bare half helix has been shortened by approximately 25%. This table is related to *Figure 7*.

| The time required to form a newly matured cofilin cluster in a pre-shortened half helix | | The duration needed to shorten a bare half helix beyond the cofilactin/bare actin boundary | | | | |
|---|---|---|---|---|---|---|
| | | P1 – P2 | P2 – P3 | P3 – P4 | P4 – P5 | P5 – P6 |
| P1 | 0 s | | | | | |
| P2 | 0 s | 0 s | | | | |
| P3 | 5.0 s | | 2.0 s (*t*=3.0 s) | | | |
| P4 | 36 s | | | 30.5 s (*t*=5.5 s) | | |
| P5 | 54.5 s | | | | 54 s (*t*=0.5 s) | |
| P6 | 58.5 s | | | | | 58.5 s (*t* = ~0 s) |

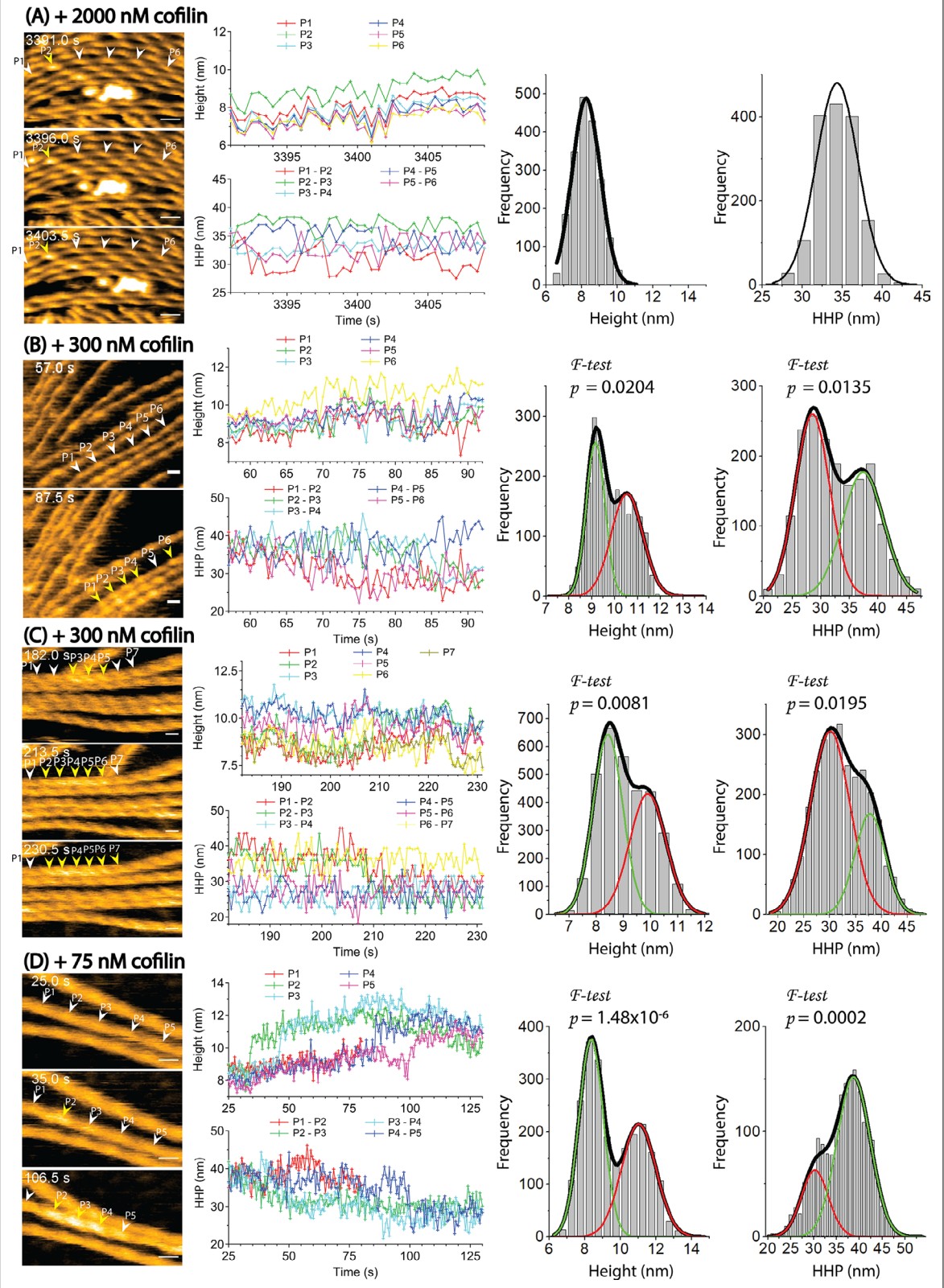

**Figure 8.** Cooperative binding of cofilin is heavily dependent on the dynamic and cooperative conformational changes involving helical twisting within actin filaments. (**A**) The cooperative binding of cofilin to actin filaments was significantly inhibited when the torsional motion during helical twisting within the filaments was suppressed. The filaments were strongly attached onto a highly positively charged lipid membrane containing 1,2-dipalmitoyl-sn-glycero-3-phosphocholine (DPPC) and 1,2-dipalmitoyl-3-trimethylammonium-propane (DPTAP) (75/25 wt%) and imaged in F1 buffer containing

*Figure 8 continued on next page*

*Figure 8 continued*

2000 nM cofilin and 0.5 mM ATP. (**B, C, D**) The normal cooperative bindings of cofilin to actin filaments were observed when the filaments were loosely immobilized on weakly positively charged lipid membrane comprising DPPC/DPTAP (90/10 wt%) and imaged in F1 buffer under different conditions: 300 nM cofilin +0.5 mM ATP (**B**), 300 nM cofilin +1 mM ADP (**C**), and 75 nM cofilin +1 mM ATP +10 mM Pi (**D**). The changes in peak heights and HHPs of a representative actin filament under each condition were analyzed over time. Histograms were constructed by analyzing the changes in peak heights and HHPs of all actin filaments in the observed field during the growth of the cofilin cluster at various time intervals after adding cofilin. The statistical significance of the peaks in the histograms was examined using an *F-test*. The green and red curves denote the fractions of F-actin and cofilactin structures, respectively. Time labels in the still AFM images indicate the elapsed time after adding cofilin to the imaging buffer. Peaks of bare actin and cofilactin are denoted by white and yellow arrowheads, respectively. Scale bars: 25 nm.

The online version of this article includes the following source data and figure supplement(s) for figure 8:

**Source data 1.** Cooperative binding of cofilin involving helical twisting within actin filaments.

**Figure supplement 1.** The spatial ACF curve analysis.

**Figure supplement 1—source data 1.** The spatial ACF curve analysis.

cleft closed and the OD untwisted in the F-actin (*Figure 1*, *Figure 2*). When cofilin binds and sits on the sides of SD1 and SD2 of two adjacent actin protomers (*Figure 2B and C*), the OD (SD1, SD2) in bound protomers becomes more twisted (a decrease in PC2), but without significantly affecting the closing of the nucleotide binding cleft (nearly unchanging PC1). This transformation ultimately leads to the conformational changes from F-actin to C-actin (*Figure 1*). The transition from the C-actin to breaking the filament into monomer G-actin structure additionally requires the opening of the nucleotide binding cleft (decreasing PC1). However, this process is normally hindered in the presence of cofilin, as the bridging between two protomers can compensate for all the losses in the disrupted interactions between the SD2 D-loop and SD1 pocket (*Figure 2B and C*). Strikingly, the HS-AFM results suggest that cofilin clusters can convert bare protomers within the shortened bare half helices on the PE side into C-actin-like structures. Unless additional cofilin molecules quickly bind to the protomers with C-actin-like structures on the PE side, the disrupted interactions would readily lead to the nucleotide binding cleft transitioning from the closed to open states, resembling C-actin to G-actin transition (*Figure 1*). Our proposed hypothesis regarding the shift from C-actin-like structure to G-actin aligns with earlier studies suggesting that the binding sites of cofilin encompass a region that overlaps with G-actin structure (*Carlier et al., 1997*; *Mannherz et al., 2007*; *Narita, 2020*; *Tanaka et al., 2018*).

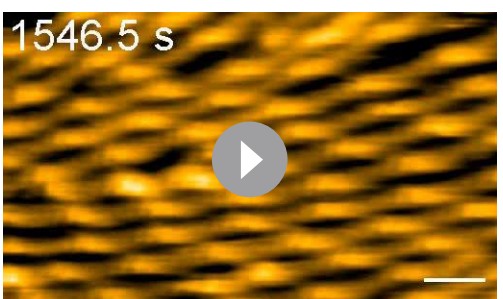

**Video 7.** Cooperative binding of cofilins along actin filaments was greatly inhibited when actin filaments were strongly immobilized on lipid membrane composed of 1,2-dipalmitoyl-sn-glycero-3-phosphocholine (DPPC) and 1,2-dipalmitoyl-3-trimethylammonium-propane (DPTAP) (75/25 wt%). The strong immobilization hindered the torsional motion of actin filaments during helical twisting, resulting in suppressing cofilin binding and cluster expansion. Actin filaments were imaged in an F1 buffer containing 2000 nM cofilin and 0.5 mM ATP. Time label denotes the elapsed time after addition of cofilin. Bars: 25 nm. Imaging rate: 2 fps. Video plays at 10 fps. Related to *Figure 8A*.

https://elifesciences.org/articles/95257/figures#video7

The dissociation of the D-loop from an adjacent actin subunit directly causes the transition to the G-form of actin, which is regarded as the most stable configuration for the actin molecule (*Oda et al., 2019*). Additionally, the protomers within the shortened bare half helix near a cofilin cluster on the PE side of F-ADP.Pi-actin may be quickly converted to F-ADP-actin and C-actin-like structure through F-actin to C-actin transition. Together, this explains the preferential binding of cofilin to ADP-actin on the PE side and why filament breakages are more commonly observed at or near the boundary of cofilactin and bare actin on the PE side, rather than where actin protomers are fully bound with cofilin (*Bibeau et al., 2021*; *Ngo et al., 2015*; *Suarez et al., 2011*). However, our finding regarding the structure partially contradicts the existing Cryo-EM structure, which suggested that conformational changes induced by cofilin clusters, shifting from F-actin to C-actin, are restricted to the protomers directly in contact with bound cofilin. According to this model, cofilin tunes adjacent protomers, resembling F-ADP-actin structure on the PE side, to expand the clusters (*Huehn et al., 2020*; *Suarez et al.,*

**Table 7.** A summary of peak heights and HHPs obtained from the histograms in *Figure 8*.

Single and double Gaussian fitting methods were applied to the data. The significant difference between the peaks in the Gaussian fitting was statistically evaluated and indicated using an *F-test*. 'ND' indicates cases where the information was not determined.

| | Peak Height (nm) | | Half Helical Pitch (nm) | | |
|---|---|---|---|---|---|
| | Peak 1 (proportion) [*F-test, p*] | Peak 2 (proportion) [*F-test, p*] | Peak 1 (proportion) [*F-test, p*] | Peak 2 (proportion) [*F-test, p*] | Frequency |
| (A) 2000 nM cofilin, 0.5 mM ATP/ on DPPC/DPTAP (75/25 wt%) | 8.2±0.8 (100%) | ND | 34.3±2.6 (100%) | ND | Height: 1930 HHP: 1556 |
| (B) 300 nM cofilin, 0.5 mM ATP/ on DPPC/DPTAP (90/10 wt%) | 9.2±0.4 (47.0%) [0.0204] | 10.5±0.7 (53.0%) [0.0204] | 28.5±3.0 (55.1%) [0.0135] | 37.4±3.6 (44.9%) [0.0135] | Height: 2212 HHP: 1780 |
| (C) 300 nM cofilin, 1 mM ADP/ on DPPC/DPTAP (90/10 wt%) | 8.4±0.5 (54.8%) [0.0081] | 9.9±0.7 (45.2%) [0.0081] | 30.1±3.8 (70.3%) [0.0195] | 37.8±2.9 (29.7%) [0.0195] | Height: 3183 HHP: 2779 |
| (D) 75 nM cofilin, 1 mM ATP, 10 mM Pi/ on DPPC/DPTAP (90/10 wt%) | 8.3±0.7 (55.3%) [$1.48 \times 10^{-6}$] | 11.0±1.0 (44.7%) [$1.48 \times 10^{-6}$] | 30.3±2.9 (25.2%) [0.0002] | 38.7±3.6 (74.8%) [0.0002] | Height: 2316 HHP: 1865 |

*2011*). Also, cofilin binding may be facilitated by a shortening of the helix pitch, independent of any transition to the C-form of actin (*Narita, 2020*). Nonetheless, their hypotheses are insufficient to explain the PCA results supporting the structural evidence obtained from our AFM measurements that a cofilin cluster not only has the capacity to convert F-actin to C-actin but also allosterically expands the C-actin structure from the bound to unbound region on the PE side. Thus, additional cofilin molecules preferentially bind to actin protomers resembling C-actin within the shortened bare zone than F-ADP-actin structure near a cofilin cluster on the PE side (*Figure 1*, *Figures 5–9*). Even though HS-AFM did not allow for the direct observation of the different nucleotide states of actin, these unidirectional propagation of the supertwisted structure in cofilin clusters to the neighboring bare zone on the PE side and preferential and cooperative bindings of cofilin to those bare sections were consistently and unequivocally observed in the presence of ADP and ADP.Pi (*Ngo et al., 2015*). Further studies are needed to identify the source of this discrepancy between HS-AFM and Cryo-EM observations.

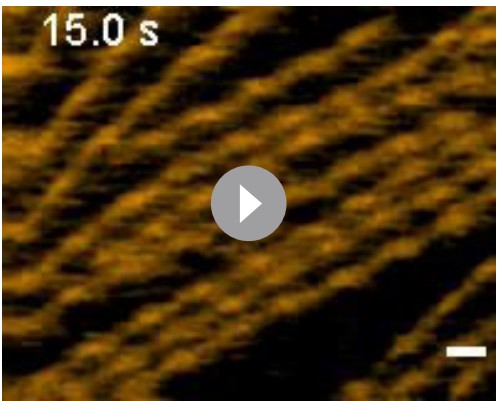

**Video 8.** Cooperative binding of cofilins along actin filaments was normal when actin filaments were loosely immobilized on lipid membrane composed of 1,2-dipalmitoyl-sn-glycero-3-phosphocholine (DPPC) and 1,2-dipalmitoyl-3-trimethylammonium-propane (DPTAP) (90/10 wt%). Actin filaments were imaged in an F1 buffer containing 300 nM cofilin and 0.5 mM ATP. Time label denotes the elapsed time after addition of cofilin. Bars: 25 nm. Imaging rate: 2 fps. Video plays at 10 fps. Related to *Figure 8B*.

https://elifesciences.org/articles/95257/figures#video8

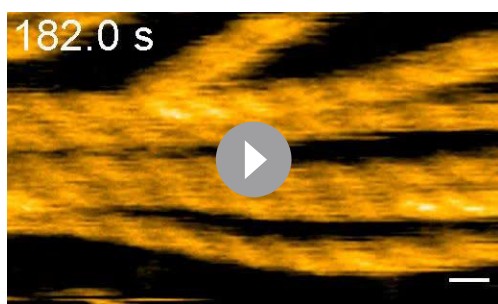

**Video 9.** Cooperative binding of cofilins along actin filaments was normal when actin filaments were loosely immobilized on lipid membrane composed of 1,2-dipalmitoyl-sn-glycero-3-phosphocholine (DPPC) and 1,2-dipalmitoyl-3-trimethylammonium-propane (DPTAP) (90/10 wt%). Actin filaments were imaged in an F1 buffer containing 300 nM cofilin, 1 mM ADP, 5 U/ml hexokinase, and 10 mM glucose. Time label denotes the elapsed time after addition of cofilin. Bars: 25 nm. Imaging rate: 2 fps. Video plays at 10 fps. Related to *Figure 8C*.

https://elifesciences.org/articles/95257/figures#video9

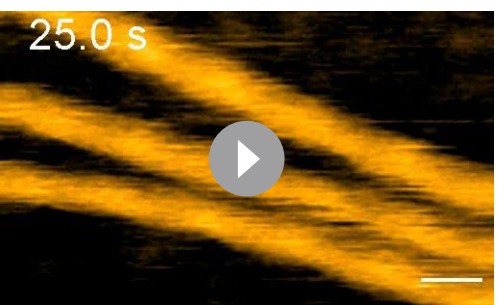

**Video 10.** Growth of cofilin cluster along actin filaments was normal when actin filaments were loosely immobilized on lipid membrane composed of 1,2-dipalmitoyl-sn-glycero-3-phosphocholine (DPPC) and 1,2-dipalmitoyl-3-trimethylammonium-propane (DPTAP) (90/10 wt%). Even with 10 mM Pi, the cofilin cluster could supertwist the half helices in the actin filaments once a critical cluster was established. Actin filaments were imaged in an F1 buffer containing 75 nM cofilin and 1 mM ATP +10 mM Pi (see Methods). Time label denotes the elapsed time after addition of cofilin. Bars: 25 nm. Imaging rate: 2 fps. Video plays at 10 fps. Related to *Figure 8D*.

https://elifesciences.org/articles/95257/figures#video10

Our aim is to understand the underlying mechanism driving the preferential and cooperative attachment of cofilin to actin filament. This mechanism is largely influenced by the preferred helical twisting structure within the actin filament, as opposed to an alternate theory that suggests cofilin binding depends on the nucleotide state of actin (*Figure 10*). While both hypotheses might explain biased cofilin binding along the actin filament, they propose different mechanisms underlying cofilin's cooperative behavior and binding preference.

Our hypothesis (*Figure 10E–H*) aligns with the results in the cofilactin region, where actin protomers are fully decorated by cofilin (*Figure 10A, C and D*). However, it is partially incongruent with the structures of actin protomers near cofilin clusters on the PE side, as depicted in *Figure 10D*. Additionally, we propose two potential hypotheses that require further verification in the near future to explain the preferential cooperative bindings and cluster expansion of cofilin to undecorated C-actin-like structure within the shortened bare half helix on the PE side (*Figure 10E–F*): (i) Because of the disrupted connections between the BE-protomer's SD2 D-loop and PE-protomer's SD1 pocket in the area resembling the C-actin structure, where cofilins are not bound, the SD2 could change between untwisted and twisted states (F-actin to C-actin transition) and the nucleotide-binding cleft could shift between the closed and open states (C-actin to G-actin transition). These transitions might prompt the rapid transformation of protomers within the shortened bare half helix adjacent to the cofilin clusters on the PE side of F-ADP.Pi-actin to F-ADP-actin and C-actin-like structure. This phenomenon also explains the binding preference of cofilin for ADP-actin on the PE side; (ii) In the canonical actin protofilament, the SD2 D-loop of BE-protomer is inserted between the tips of SD1 and SD3 of PE-protomer (*Figure 2A*). However, the SD2 D-loop of BE protomer does not directly interact with the tips of SD1 and SD3. Instead, it is deeply buried within the core of the cofilactin protofilament, representing the C-actin structure (*Figure 2B–C*). Conversely, in the region representing the C-actin-like structure, where cofilins are unbound, the SD2 D-loops in twisted OD might exhibit various dynamic conformations, potentially allowing flexible interaction with the tips of SD1 and SD3 including that of the neighboring protomers on the PE side. The collective movements of individual ODs that decrease PC2 values would lead to a shorter HHP and fewer number of protomer pairs per HHP (*Figure 2*). These changes cause an extended MAD in C-actin-like structure area with optimal conformational changes in lower cleft between SD1 and SD3 of the protomers on the PE side, thereby facilitating the preferential binding and expansion of cofilin clusters (*Figure 10E–F*).

In this study, we experimentally demonstrated that helical twisting, the number of protomers per half helix, and MADs are intrinsically dynamic and fluctuate around their mean values over time. Our findings support the variability (*Egelman et al., 1982*) and irregularity (*Fineberg et al., 2024*) in helical twists and dynamics of actin filaments both in vitro and in vivo (*Ivanova et al., 2024*), challenging the traditional view, based on Cryo-EM and X-ray diffraction, that these parameters remain unchanged. Furthermore, we observed a significant correlation between the nucleotide-bound states of actin and the proportion of naturally supertwisted half helices in F-ADP-actin and F-ADP.Pi-actin (*Figure 9*). We suggest that the bound Pi reduces the fraction of naturally supertwisted half helices and hinders the initial binding of cofilin and/or initial growth to a critical size of cluster set at two to four molecules (*Figure 5*, *Figure 6*) that can induce allosteric conformational changes to the neighboring bare zone on the PE side for cooperative bindings of cofilin (*Figure 9C–D*, *Figure 10H*). This discovery aligns with previous research, which showed that achieving the most effective cooperative binding of cofilin

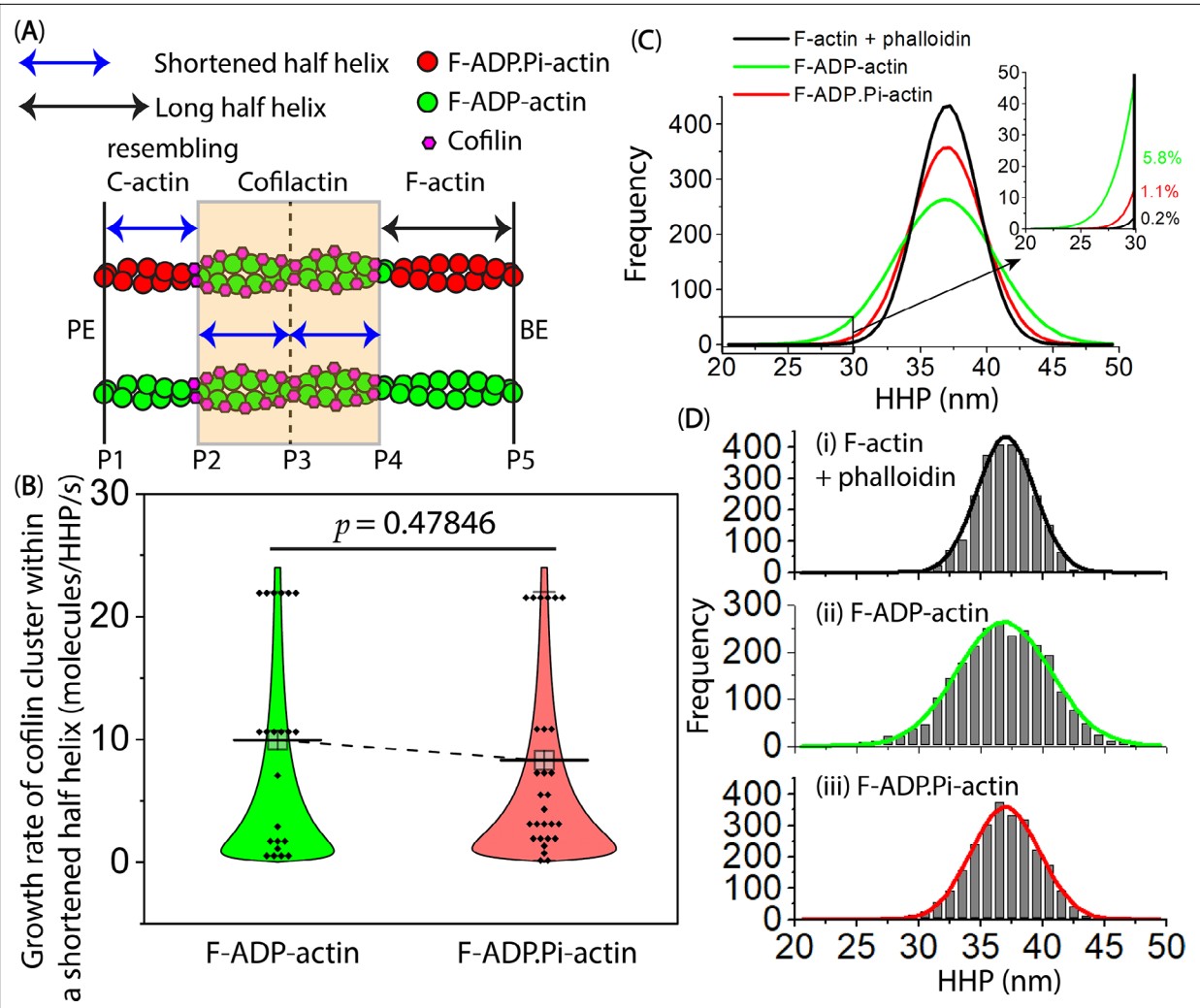

**Figure 9.** Comparison of the growth rate of cofilin clusters within the shortened bare half helices that resemble the C-actin structure adjacent to cofilin clusters on the PE side, for both F-ADP-actin and F-ADP. Pi-actin.(**A**) In this cartoon representation, hybrid cofilactin/bare actin filaments are depicted, consisting of four half helices. Two of these half helices are fully decorated with cofilin clusters in the middle, followed by a shortened bare half helix and a normal long bare half helix on the PE and BE sides, respectively. P1-P5 were labels indicating peak heights at individual crossover points. To make a meaningful comparison, the lag time measurements were initiated exclusively when a bare half helix between P1 and P2 experienced shortening, and they continued until a newly matured cluster emerged in P1 within the shortened bare half helix (**P1–P2**) adjacent to the preformed cofilin cluster (**P2**). Each fully decorated supertwisted half helix contains 11 cofilin molecules (referred to *Figure 2—figure supplement 1I*). To determine the mean growth rate of the cofilin cluster within a shortened bare half helix adjacent to the cofilin cluster, 11, the number of cofilin molecules in one half helix, was divided by the lag time in seconds. (**B**) This part presents a comparison of the growth rate of cofilin clusters within shortened bare half helices that resemble the C-actin structure adjacent to cofilin clusters on the PE side. This comparison was made when actin filaments were incubated with the same cofilin concentration (i.e. 300 nM) in the F1 buffer containing 1 mM ADP +5 U/ml hexokinase +10 mM glucose or 1 mM ATP +10 mM Pi, resulting in filamentous protomers representing F-ADP-actin or F-ADP.Pi-actin structures. The growth rates (mean ± SD) were measured for 22 shortened bare half helices in F-ADP-actin (10.0±8.3 molecules/HHP/s) and 28 shortened bare half helices in F-ADP.Pi-actin (8.3±7.7 molecules/HHP/s). Using the two-population *t-test*, the results showed that the difference was not significant (p=0.47846) at the p≤0.05 level. (**C, D**) Histograms illustrate the distribution of the HHP (mean ± SD) in control actin filaments for three conditions: (**i**) phalloidin-stabilized F-actin (37.0±2.3 nm), (**ii**) F-ADP-actin (36.9±3.8 nm), and (**iii**) F-ADP.Pi-actin (37.0±2.8 nm), with each condition having 2487 data points. For comparison, single Gaussian fits in D were integrated and displayed in C. The inset within C exhibited fractions of the naturally supertwisted half helices with HHPs shorter than 30 nm. Utilizing one-way ANOVA with a significance level of p≤0.05, there were significant differences observed between (**i**) and (**ii**) (p=0.00837, F=6.95754) as well as between (**ii**) and (**iii**) (p=0.04080, F=4.18633), primarily attributable to the statistical difference in the variance of HHP between condition (**ii**) vs. (**i**) and (**iii**). The difference observed between (**i**) and (**iii**) was not significant (p=0.58272, F=0.30189).

The online version of this article includes the following source data for figure 9:

**Source data 1.** The growth rate of cofilin clusters within the shortened bare half helices for both F-ADP-actin and F-ADP. Pi-actin.

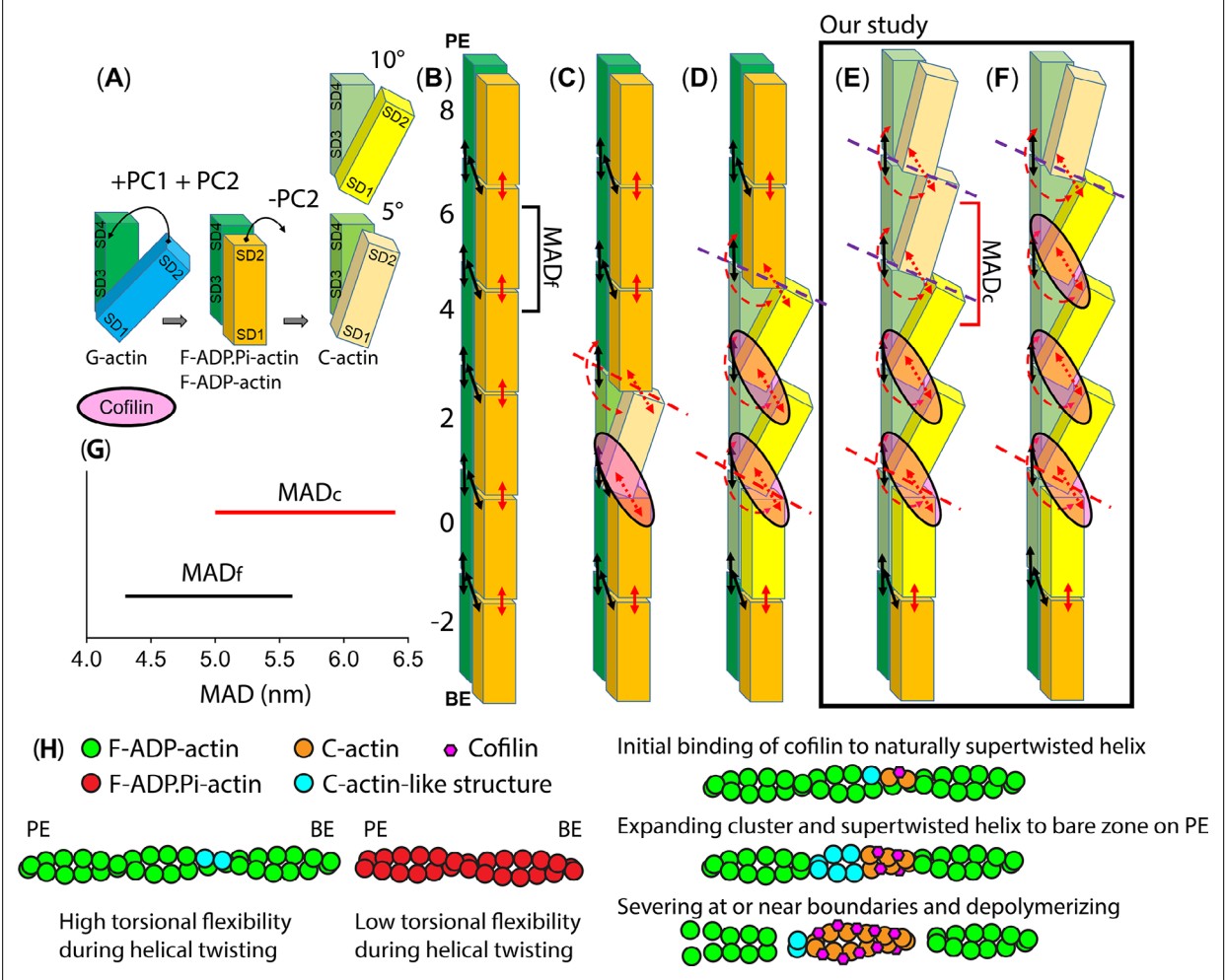

**Figure 10.** The hypothesis regarding the actin structure-dependent preferential cooperative binding of cofilin. For simplicity, a single actin protofilament comprising six protomers labelled from –2 to 8 is shown, both without and with cofilin. The black and red arrows represent stable bonds and unstable bonds between domains of two adjacent actin protomers in the same strand, respectively. The broken red arrows indicate completely disrupted bonds between domains of two adjacent actin molecules. Additionally, broken red and purple lines show positions sensitive to being severed, with the purple lines indicating a stronger severing activity. (**A**) The conformational transition between G-actin, F-actin (bound with ADP.Pi, ADP), and C-actin structures. In the lower model of C-actin, the binding of a single cofilin molecule to an actin filament induces a small movement of SD2, causing a rotational shift of approximately 5° in the OD relative to the ID. In the upper one, when the cofilin clusters bind, they induce a larger rotation of approximately 10° in the OD (*Galkin et al., 2011*; *Tanaka et al., 2018*). (**B**) A single protofilament of an actin filament. Upon polymerization, OD rotates relative to the ID to close nucleotide binding cleft and untwist the OD to flatten actin molecule. The orange color corresponds to OD (SD1 + SD2), while green corresponds to ID (SD3 + SD4; *Oda et al., 2009*). (**C**) A single protofilament of an actin filament bound to a single cofilin (*Huehn et al., 2020*; *Tanaka et al., 2018*). (**D**) A single protofilament of an actin filament bound to cofilin clusters. Actin conformational changes induced by cofilin clusters are local and limited to protomers directly contacting bound cofilin. The cofilin bindings accelerate a rapid transition from F-ATP-actin to F-ADP.Pi-actin and F-ADP-actin states of protomers on the PE side. Cofilin tunes adjacent protomers, resembling F-ADP-actin structure on the PE side, to expand the clusters. The cooperative binding of cofilin requires a critical size of a cofilin cluster set at 2. Notably, the severing of the actin filament is predominantly observed at the junction between cofilactin and bare actin (*Huehn et al., 2020*; *Suarez et al., 2011*; *Tanaka et al., 2018*). (**E–F**) A single protofilament of an actin filament bound to cofilin clusters in the authors' previous study (*Ngo et al., 2015*) and in the current study. Our hypothesis supports that actin conformational changes induced by cofilin clusters are not local and not limited to protomers directly contacting bound cofilin (C-actin) on the PE side. Cofilin clusters have the ability to shorten a half helix by reducing the number of protomers. Additionally, cofilin cluster not only has the capacity to convert F-actin to C-actin within cofilactin region but also allosterically expands the C-actin structure from the bound to unbound region on the PE side (**E**). The MAD$_c$ in the undecorated C-actin-like structure region are elongated compared to MAD$_f$ in F-actin region. The shortening of half helices and the elongated MAD$_c$ play a pivotal role in facilitating the preferential cooperative binding of cofilin to actin protomers via F-actin to C-actin transition pathway. If cofilin does not promptly attach to protomers in shortened bare half helix region (as depicted in E-F), the protomers within C-actin-like structure regions can readily open the nucleotide binding clefts and transition back to G-actin state via C-actin to G-actin transition, ultimately severing the filament at or near boundaries (broken lines). (**G**) HS-AFM measurements of MAD values measured in half helices of filaments representing F-actin (MAD$_f$) and undecorated C-actin-like structure (MAD$_c$) adjacent to cofilin cluster on the PE side. (**H**) A hypothesis describing how the function of cofilin

*Figure 10 continued on next page*

*Figure 10 continued*

is regulated by the torsional flexibility during helical twisting of actin filaments. The F-ADP-actin has increased the fractions of naturally supertwisted helical structures and the torsional flexibility during helical twisting, thereby creating some proposed C-actin-like structures to facilitate the initial binding and cluster expansion of cofilin, compared to F-ADP.Pi-actin. To emphasize the inhibitory effect of Pi on the initial binding of cofilin, we depict a simplified model of F-ADP.Pi-actin, featuring very infrequent structures resembling C-actin. As a result, the initial binding of cofilin and growing of critical cluster set at 2–4 cofilin molecules are more challenging compared to F-ADP-actin.

also requires a critical size for a cofilin cluster set at two molecules (*Huehn et al., 2020*). However, Pi only minimally impedes the propagation of the supertwisted helical structure within the cofilin cluster to the bare actin zone in F-ADP.Pi-actin compared to F-ADP-actin. This transition is pivotal in driving the preferential cooperative binding and expansion of the cofilin cluster (*Figure 7*, *Figure 8*). In terms of structure, the PCA results also strongly agree with previous research indicating that ADP.BeF₃, ADP.Pi, and ADP F-actin structures (*Figure 1—figure supplement 1*) display a striking resemblance (*Oosterheert et al., 2022*; *Reynolds et al., 2022*). Additionally, the reduction in torsional flexibility during helical twisting resulted in a significant hindrance to the cooperative binding and expansion of cofilin clusters (*Figure 8A*). Therefore, we suggest that the reduction in torsional flexibility associated with variable helical twists, rather than the large conformational change in F-actin's protomers linked to ATP hydrolysis, accounts for the inhibition of cofilin binding by Pi (*Figure 10H*). Nonetheless, it remains plausible that the structural flexibility exhibited by ADP-bound actin protomers could result in subtle variations in the conformations of the DNase binding loop (D-loop) G46-M47-G48-N49, as suggested in *Chou and Pollard, 2019*. The crystal structure of the F-form revealed that Pi in ADP.Pi connects the two large domains of the actin molecule, thereby stabilizing the F-form. When Pi is released, this connection is significantly weakened (*Kanematsu et al., 2022*). We suggest that the absence of bound Pi possibly increases the torsional flexibilities during helical twisting of ADP-bound actin filaments in contrast to their ADP.Pi-bound counterparts.

Actin structure-dependent preferential binding of actin-binding proteins (ABPs) have been investigated at the molecular and cellular levels (*Hayakawa et al., 2023*; *Jegou and Romet-Lemonne, 2020*; *Jégou and Romet-Lemonne, 2016*; *Ngo et al., 2016*; *Tokuraku et al., 2020*; *Tokuraku et al., 2009*). Notably, Harris and colleagues reported biased localization of tandem calponin homology domains (CH1-CH2) that can be mutated to selectively bind different actin conformations at the leading or trailing edge of motile cells (*Harris et al., 2020*). The untwisting of the long-pitch helix of the actin filament induced by diaphanous-related formin mDia1 inhibits cofilin binding (*Mizuno et al., 2018*). Stretching actin filaments within cells enhances their binding affinities for the myosin II motor domain (*Uyeda et al., 2011*). The importance of dynamic, cooperative, and allosteric conformational changes in actin filaments warrants further investigation of the actin structure-dependent preferential cooperative binding and function of other ABPs to clarify the role of dynamics, structural polymorphisms, and physiological implications of actin filaments in working with different ABPs in cells.

## Methods

### Proteins

Rabbit skeletal actin and human cofilin 1, without a His-tag, were prepared following the same protocol as described in our previous report (*Ngo et al., 2015*). The purified proteins were rapidly frozen in liquid nitrogen and stored at –80 °C until use. Prior to experimentation, the proteins were thawed on ice and subjected to ultracentrifugation (typically at 80,000 rpm for 10 min at 4 °C) to eliminate any undesired proteins present in the pellet. The remaining high-quality proteins in the supernatant were collected, kept on ice, and utilized for HS-AFM experiments.

### Principal component analysis

Actin molecules have been categorized into four groups based on the orientations of the core of the outer domain (OD): F-form in naked F-actin, C-form in cofilactin, O-form in profilin-actin, and G-form in presumably free actin, as determined by Hierarchical Cluster Analysis (HCA) (*Oda et al., 2019*). However, HCA serves only for clustering and lacks the capability to clearly visualize or map back to the original structural data to derive further meaningful insights. In contrast, Principal Component Analysis (PCA) is typically used to reduce the dimensionality of large datasets, retaining essential

information. Through this dimension reduction, data clusters naturally form, and critical information can be mapped back to the original data, revealing collective movements between clusters. PCA has previously been applied to analyze actin structures (*Xue et al., 2014*) using a small set of PDBs.

The process involves aligning the original data in three-dimensional space to remove trivial movements, calculating the displacement covariance matrix between common atoms, and performing eigendecomposition. This results in a small number of new variables called principal components (PCs), which capture most of the original data's information. Clustering is a byproduct of PCA, as the original dataset can be projected onto the identified collective components and naturally grouped in the new PC dimensional space.

To ensure consistent and accurate PCA results, we conducted several preprocessing steps. After cleaning the selected PDBs, we specifically extracted the alpha carbons of all reported protein chains for further analysis. The sequences of these chains were aligned using the AlignIO module from the Biopython package (*Cock et al., 2009*), and we identified the residues that were commonly present across the PDBs. Specifically, we found that residues 8–41, 54–63, 66–74, 76–234, and 236–373 in the ACTs_RABIT gene (P68135) were common among the 46 chosen PDBs. Importantly, these residue numbers differ from the standard residue IDs used in PDBs because the initial two residues present in genes are typically cleaved in the final actin chains.

We then performed structural alignment only for residues in the IDs, following the definition reported in *Kabsch et al., 1990*. After aligning the structures, we used the entire set of commonly solved residues to establish the covariance matrix, which was subjected to eigendecomposition to obtain different eigenvectors (PCs). As shown in *Figure 1* and *Figure 1—figure supplement 1*, we utilized the two eigenvectors with the largest values, denoted as PC1 and PC2. In this study, the significance of PCA lies in its capacity to unveil the conformational transition between G-actin and C-actin, and vice versa, through PC1, while PC2 delineates the pathway between F-actin and C-actin, and vice versa.

## Analyses of pseudo AFM images of F-actin and C-actin structures constructed from existing PDB structures

To illustrate how cofilin clusters impact the helical twisting of long actin filaments, leading to the shortening of HHPs and less protomers per HHP, resembling the C-actin structure as measured by AFM, we built long actin filaments at different states along the F-actin to C-actin transitions by using 8 PDB chains (*Table 2*). In *Figure 2—figure supplement 1*, we show pseudo-AFM images featuring long filaments representing F-actin (constructed from 6VAU), C-actin (with cofilins excluded) and cofilactin (constructed from 6VAO) structures.

To construct the long filament featuring F-actin, we performed a series of steps. First, we repeatedly rotated and translated two actin chains from the selected PDBs to generate the coordinates of 62 consecutive protomers. Since not all of the selected PDBs had reported BIOMT matrices, we used transformation matrices that minimized the root mean square deviation (RMSD) between the two chains. To construct long filaments featuring C-actin and cofilactin, a similar process was followed, but it involved two chains of actin protomers and two chains of cofilins. The lists of chains and matrices utilized to build the long filaments are presented in *Table 2*. Zoom-in views of these long filaments are shown in *Figure 2*.

The generated long filaments were then aligned along the x-axis and pseudo-imaged using Afmize software (https://github.com/ToruNiina/afmize; *Niina, 2024*; *Niina et al., 2021*; *Niina et al., 2020*). A probe radius of 2.0 nm with an angle of 5.0° was set, and the image resolution was configured to be 1.0 nm in the xy plane and 0.64 Å on the z-axis, with a noise level of 0.1 nm. To analyze the structures, we calculated longitudinal section profiles of the HHPs by determining the average height within ±3 nm around the center line (red and blue lines in *Figure 2—figure supplement 1A*). The offset values of the height profile were set to zero by referring to virtual mica surface, where filament was placed in the pseudo-AFM images.

In our AFM analyses, we determined the HHPs by measuring the distances between major peaks observed in longitudinal section profiles along the filament axis (red markers in *Figure 2—figure supplement 1A,B,D,E,G,H*). These major peaks likely represent two distinct actin protofilaments that are successively probed by the AFM tips (red markers in *Figure 2—figure supplement 1C,F,I*). Thus, the number of actin protomers within each HHP is odd, leading to a noninteger number of actin pairs.

For example, in canonical actin filaments, each HHP is typically composed of 13 actin protomers (equivalent to 6.5 protomer pairs). To account for this, we determined the number of protomer pairs within each HHP by counting the major and minor peaks (noted by the red and green markers within each HHP subtracted by 0.5) in the same longitudinal section profiles. This adjustment was applied for both the pseudo and experimental AFM images of the filaments representing F-actin and C-actin. Our pseudo-AFM image displayed the challenge of accurately detecting actin pairs within filaments representing the cofilactin structure (*Figure 2—figure supplement 1G–I*) due to the presence of cofilin decorations and the tip-sample dilation effect (*Ando, 2022*).

## High-speed atomic force microscopy

We used a laboratory-built high-speed atomic force microscope (HS-AFM), as described previously (*Ando et al., 2013*). HS-AFM imaging in the tapping mode was carried out in solution with small cantilevers (BL-AC10DS-A2, Olympus), and the spring constant, resonant frequency in water, and quality factor in water were ~0.1 N/m, ~500 kHz, and ~1.5, respectively. An additional tip was grown on the original cantilever tip in gas with sublimable ferrocene powder by electron beam deposition (EBD) using scanning electron microscopy (ZEISS Supra 40 VP/Gemini column, Zeiss, Jena, Germany). Typically, the EBD tip was grown under vacuum ($1–5\times10^{-6}$ Torr), with an aperture size of 10 μm and an electron beam voltage of 20 kV for 30 s. The EBD ferrocene tip was sharpened using a radio frequency plasma etcher (Tergeo Plasma Cleaner, Pie Scientific, Union City, CA) under an argon gas atmosphere (typically at 180 mTorr and 20 W for 90 s). For HS-AFM imaging, the free-oscillation peak-to-peak amplitude of the cantilever ($A_0$) was set at ~1.6–1.8 nm, and the feedback amplitude set-point was set at ~0.9 $A_0$. HS-AFM imaging was performed as described in detail elsewhere (*Ando et al., 2013*; *Kodera et al., 2021*; *Ngo et al., 2015*), except for the use of a recently developed only tracing imaging (OTI) mode (*Fukuda and Ando, 2021*).

## HS-AFM imaging

The lipid compositions used in this study predominantly consisted of neutral 1,2-dipalmitoyl-*sn*-glyc ero-3-phosphocholine (DPPC, Avanti) and positively charged 1,2-dipalmitoyl-3-trimethylammonium-propane (DPTAP, Avanti). Liposomes were prepared using DPPC/DPTAP (90/10, wt%), DPPC/DPTAP (75/25, wt%), and DPPC/DPTAP (50/50, wt%) ratios, following our established sample preparation protocol (*Ngo et al., 2015*). These liposomes, along with mica-supported lipid bilayers, were created accordingly.

Actin filaments were generated in F1 buffer containing 40 mM KCl, 20 mM PIPES-KOH (pH 6.8), 1 mM $MgCl_2$, 0.5 mM EGTA, and 0.5 mM DTT in the presence of 0.5–1 mM ATP, following the procedure described in our previous work (*Ngo et al., 2015*). To obtain actin filaments bound with different nucleotides (ATP, ADP, ADP.Pi) and investigate the cooperative binding of cofilin to these filaments, a similar protocol was employed (*Ngo et al., 2015*). Briefly, actin filaments (5–10 μM) were made in F1 buffer containing 0.5–1 mM ATP and gently immobilized on a positively charged lipid membrane composed of DPPC/DPTAP (90/10 wt%). The actin filaments on the lipid surface were washed with 20 μl of imaging F1 buffer additionally including either 0.5 mM ATP, 1 mM ADP/5 U/ml hexoki-nase/10 mM glucose, or 1 mM ATP +10 mM Pi ($K_2HPO_4$/$KH_2PO_4$ (pH 6.8) buffer), resulting in F-ATP-actin, F-ADP-actin, or F-ADP.Pi-actin, respectively. Subsequently, cofilin, prepared in the imaging F1 buffer with the desired concentration and nucleotides, was introduced into the imaging chamber to observe the cooperative binding of cofilin along the actin filaments.

To compare the helical twisting structures, the number of protomers per HHP, and MADs in the normal F-actin structure with that in bare actin segments on both sides of the cofilin clusters in the hybrid cofilactin/bare actin filaments at high spatiotemporal resolution in the same imaging field, we immobilized these filaments sequentially on the same lipid membrane (referred to *Figure 5—figure supplements 1–2*). First, actin filaments were prepared by incubating G-actin in F1 buffer and 0.5 mM ATP at a concentration of 20 μM for approximately 1 hr at 22 °C, resulting in F-actin. Second, cofilactin and hybrid cofilactin/bare actin filaments were formed by mixing F-actin (10 μM) with cofilin (1 μM) in a small tube in F1 buffer and 0.5 mM ATP. The final molar ratio was 10/1 (actin to cofilin). The mixture was incubated for 10 min at 22 °C to allow the binding, cluster expansion and actin filament severing to reach equilibrium. Third, a 2 μl solution containing cofilactin and hybrid cofilactin/bare actin fila-ments was introduced onto a lipid membrane composed of DPPC/DPTAP (50/50 wt%) and incubated

for 10 min at 22 °C to immobilize the filaments. Fourth, the surface was washed with 20 µl of imaging F1 buffer containing 0.5 mM ATP, and then a 2 µl solution containing normal actin filaments (10 µM) was introduced onto the same lipid membrane. This sample was also incubated for 10 min at 22 °C. Fifth, the surface was washed again with 20 µl of imaging F1 buffer containing 0.5 mM ATP prior to HS-AFM imaging.

## Analysis of the growth rate of the cofilin cluster within the shortened bare half helices adjacent to the cofilin cluster in F-ADP.Pi-actin and F-ADP-actin

We measured the growth rates of the cofilin clusters within shortened bare half helices adjacent to the cofilin clusters on the PE side in F-ADP.Pi-actin and F-ADP-actin. As noted in the section of Results, when we were able to observe bare sections on both sides of a cofilin cluster, one of them had shorter HHP while the other was close to normal HHP, and based on our previous report (*Ngo et al., 2015*), we judged the bare section with shorter HHP is on the PE side of an actin filament. It is important to clarify that the collected data did not include the growth rates of cofilin within the normal half helices on the PE or BE sides, in order to exclude the random binding events of cofilin to actin filaments under various nucleotide conditions. By employing HS-AFM, we were able to monitor changes in the length of a half helix adjacent to a cofilin cluster and the peak height at individual crossover points. The lag time measurements were performed at the precise moment when the bare half helix adjacent to the cofilin cluster on the PE side began to shorten and continued until a newly saturated cluster emerged within this neighboring bare half helix. This determination was made by analyzing the increase in the peak heights (P1) before and after forming a newly matured cofilin cluster adjacent to a preformed cofilin cluster (P2) (as depicted in *Figure 9A*). The data presented in *Figure 7* and *Figure 8C* were also included in this analysis. Each fully decorated-supertwisted half helix consists of 11 cofilin molecules (referred to *Figure 2—figure supplement 1I*). To calculate the mean growth rate of the cofilin cluster within a shortened bare half helix adjacent to a cofilin cluster, we divided 11 cofilin molecules by the lag time in seconds.

We assessed torsional flexibility of actin filaments by measuring the HHPs in control actin filaments incubated with 1 mM ADP, 1 mM ATP +10 mM Pi, or 1 mM ATP +phalloidin saturated to actin, resulting in F-ADP-actin, F-ADP.Pi-actin, or phalloidin-stabilized F-actin, respectively, using our previous method (*Ngo et al., 2015*). Because, we did not examine the nucleotide-bound actin states when actin filaments were incubated with 1 mM ATP and phalloidin excess, we simply labelled them as phalloidin-stabilized F-actin and used as the control. The fraction of naturally supertwisted half-helices with HHPs less than 30 nm was determined by computing the areas under the curves derived from single Gaussian fitting curves (referred to *Figure 9C–D*).

## HS-AFM data analysis and processing

The AFM data were analyzed using custom software, including Kodec, UMEX Viewer, and Umex Viewer for Drift Analysis, as described in previous studies (*Hayakawa et al., 2023*; *Ngo et al., 2015*). To measure the half helical pitch (HHP) and the number of actin protomer pairs per HHP, we utilized the custom software Umex Viewer for Drift Analysis, as described in *Hayakawa et al., 2023*. Additionally, AFM images and videos were processed using Fiji-ImageJ (NIH, USA) and After Effects (Adobe, USA).

To quantitatively analyze the HHPs and count the number of protomer pairs per HHP, we employed Umex Viewer for Drift Analysis, which enabled the semiautomatic determination and measurement of the distance between the highest 'cross-over' points between two long-pitch strands. This involved creating a topographical line profile along the actin filaments, similar to previous studies conducted by *Amyot et al., 2022*; *Hayakawa et al., 2023*. Specifically, one HHP was measured between the two highest cross-over points, while one helical pitch (HP) was measured between the three highest cross-over points. In brief, we drew a longitudinal section profile line along the actin filament, covering a length of 1–4 consecutive HHPs. The offset values were set to zero with reference to the height of the lowest position along the longitudinal section lines. Before analysis, we corrected the nonlinearity of the XY piezos by applying nonlinear image scaling and reduced image noise using a Gaussian smoothing filter with a standard deviation of 0.76 nm. The profile was obtained by averaging the signal within a 5–8 nm bandpass filter along the filament. To ensure correct detection of axial distance

(AD) between two adjacent protomers within the same long-pitch strand, we set a range of 3–6.6 nm for minor pitches. To facilitate correct detection of the HHPs between two crossover points, we set a range of 15–25 nm for minimum major pitches for cofilactin and bare actin segments. These allow the software to automatically penalize incorrect peaks. In some rare cases, the analyst also made the decisions to add missing peaks or to remove incorrect peaks caused by noise. Typically, we selected 30–61 actin filaments and measured the HHPs for approximately 20 consecutive time frames. To minimize measurement errors in the HHP and the number of protomer pairs per HHP, we specifically chose straight actin segments with clearly discernible protomers and drew line profiles (with an average width of 3%) to measure 1–4 consecutive HHPs. This method was consistently applied to analyze normal bare actin, cofilactin, and hybrid cofilactin/bare actin filaments. We distinguished normal F-actin and cofilactin structures based on differences in the peak height and HHP, as described in our previous study (*Ngo et al., 2015*).

To calculate the number of protomer pairs per HHP and the mean axial distance (MAD) between two adjacent protomers along the same long-pitch strand, we semiautomatically detected AD and counted the number of protomer pairs per HHP, as mentioned in other parts and shown in *Figure 2—figure supplement 1*. The MADs within an HHP were determined by dividing the HHP by the actual counted number of protomer pairs per HHP.

To enhance the characteristics of actin protomers in the different types of filaments (bare F-actin, cofilactin and C-actin-like structure in the hybrid cofilactin/bare actin segments), we converted the raw AFM images into Laplacian-filtered images. This was achieved by applying a high-pass filter known as the Laplacian of Gaussian (LoG) filter, which can be found at the following link: (https://homepages.inf.ed.ac.uk/rbf/HIPR2/log.htm). The Laplacian filter used in this process is a type of high-pass filter, characterized by the kernel $\begin{bmatrix} -1 & -1 & -1 \\ -1 & 8 & -1 \\ -1 & -1 & -1 \end{bmatrix}$. Prior to the application of the Laplacian filter, we employed a Gaussian filter with a typical sigma value of 1.27 pixel/nm. This preliminary step helped reduce sensitivity to noise and enhanced the resulting Laplacian-filtered images, resulting in improved visualization of the actin protomers.

## Temporal autocorrelation analysis

The temporal autocorrelation analysis method can be found at the following link: here.

In summary, the analysis involves the following steps: (i) Data processing and collection: We directly measured and collected time series data for the axial distance (AD) between two adjacent actin protomers along the same long-pitch strand in different actin filaments over time, as shown in *Figure 3D*, using a custom semiautomatic analysis software called Umex View for Drift Analysis. First, we converted raw AFM images containing actin filaments into Laplacian-filtered images by applying a Laplacian of Gaussian (LoG) filter (typically with sigma ~1.27 pixel/nm) to eliminate low-frequency and high-frequency noise signals. Second, we selected a short segment of the actin filament (~4 HHPs) located at the center of the AFM image and used a segmented line running along the axis of the actin filament to measure the distance between actin protomer pairs, as depicted in *Figure 3D*. To facilitate the semiautomatic analysis and determine the peak-to-peak distance between individual ADs, we applied a bandpass filter to remove noise by setting the low-pass frequency (LPF) cutoff at 3 nm and the high-pass frequency (HPF) cutoff at 6 nm for all analyses. To ensure correct detection of AD between two adjacent protomers within the same long-pitch strand, we set a range of 3–6.6 nm for minor pitches, allowing the software to penalize incorrect peaks. Typically, the variation in the AD within each actin filament over time was assessed using a series of AFM images (including 258 images captured at a rate of 2 frames per second (fps)). (ii) Calculation of the autocorrelation coefficients: Each autocovariance value was divided by the autocovariance at lag 0 to obtain the autocorrelation coefficients. These coefficients represent the strength and direction of the linear relationship between observations at different time lags (*Equation 1*). (iii) Plotting the autocorrelation function (ACF): The autocorrelation coefficients were visualized by plotting the autocorrelation function (ACF). The ACF plot displays the autocorrelation coefficients on the y-axis and the time lags on the x-axis, as depicted in *Figure 3E*. (iv) Estimation of the time constants: The ACF plot was fitted with a second-order exponential decay function to extract the time constants ($t_1$, $t_2$), as depicted in *Figure 3E–F*.

The *ACF* (*k*) of a time series of values ($y_i$) at lag *k* is defined as

$$ACF(k) = [\frac{1}{n}\sum_{i=1}^{n-k}(y_i - \bar{y})(y_{i+k} - \bar{y})]/[\sum_{i=n}^{n}(y_i - \bar{y})^2] \tag{1}$$

In our analysis, a time series of values ($y_i$) of the axial distance between two adjacent actin pairs at lag $k$ ($k{\geq}0$) was measured for each actin filament in 258 consecutive AFM images (n=258) collected at a frame rate of 2 fps, or 5 fps and 10 fps (*Figure 3—figure supplement 1*). The mean value, $\bar{y}$, was calculated.

## Spatial autocorrelation analysis

To evaluate the degree of spatial clustering in the actin protomer structural data, we conducted spatial autocorrelation analysis, as depicted in *Figure 4—figure supplement 1*. Initially, we generated a line profile by utilizing cubic spline interpolation along the ridgeline of an actin filament (*Figure 4—figure supplement 1A*). The actin filament includes a large helical twist and protomer structures with periodicities of 36 nm and 6–7 nm, respectively. The helical twist exhibits a typical height of 3 nm (*Figure 4—figure supplement 1B*), which is three times larger than the 0.5–1.5 nm height of the protomer (*Figure 4—figure supplement 1C*). To explore the spatial self-correlation among protomers, we applied a fast Fourier transform (FFT) and utilized a low-pass filter (LPF) and high-pass filter (HPF). The filters isolated heights using LPF and HPF within a range of 5–8 nm in the Fourier domain, corresponding to the actin protomer height along the filament (depicted in *Figure 4—figure supplement 1C*). To ensure correct detection of AD between two adjacent protomers within the same long-pitch strand, we set a range of 3–6.6 nm for minor pitches, allowing the software to penalize incorrect peaks. Then, we derived the autocorrelation function (ACF) of the protomer heights by obtaining the complex conjugate (magnitude) and then performed an inverse FFT. Additionally, we applied the Hilbert transform in the Fourier domain by doubling the positive frequency data, zeroing the negative frequency data, and applying an inverse FFT. This process produced an envelope curve. As a result, the envelop curves of the height profiles with and without the BPF exhibit alternating positive and negative values at half helical pitch and protomer periodicities, respectively (*Figure 4—figure supplement 1D–E*). In contrast, the envelope curve consistently shows positive values; thus, this curve is suitable for evaluating spatial autocorrelations. In the main text, we compared the envelope curves representing the spatial ACF results, measured at multiple time points and different conditions.

By visualizing the spatial ACF curves with distance ($d$) derived from F-actin and hybrid cofilactin/bare actin segments (*Figures 4C and 6C* and *Figure 4—figure supplement 2A–B*), along with sparsely decorated cofilin-actin filaments strongly attached on lipid surface (*Figure 8—figure supplement 1*), we predicted that the curves could be modelled using an exponential decaying sinusoidal function. This function, characterized by two primary components (exponential decay and sinusoid frequencies), enables us to derive parameters for the decay constant ($D1$) and the frequencies of the sinusoidal oscillation ($D2$, $D3$, …$D9$), taking into account the phase angle of the sinusoidal waves ($\gamma1$, $\gamma2$, $\gamma3$, $\gamma4$). Based on previous studies (*Ju and Rappaport, 2021*; *Zhang et al., 2008*), we successfully formulated and applied a novel exponential decaying sinusoidal function, denoted as $\rho(d)$, to fit the spatial ACF curves (*Equation 2*).

$$\rho(d) = exp(-d/D1)\,[D2\cos(d/D3 + \gamma1) + D4\sin(d/D5 + \gamma2) + D6\cos(d/D7 + \gamma3) + D8\sin(d/D9 + \gamma4)] \tag{2}$$

For fitting, we utilized the Nadam optimizer (*Dozat, 2016*) and employed gradient descent in our approach. The mean square error served as our selected loss function for fine-tuning parameters. This optimization framework not only aided in the model's convergence but also improved its capacity to precisely capture the inherent patterns in the data. Consequently, it enhanced the effectiveness of our fitting solution. Finally, the $D1$ values, providing an indirect measure of the degree to which height values near each other in space are similar to one another in actin filaments, were shown.

## Acknowledgements

The authors express their gratitude for the facility and financial support provided by the WPI Nano Life Science Institute (WPI NanoLSI), Kanazawa University, as well as the instrumental assistance provided by Toshio Ando. Special thanks are extended to Huan Tran from Georgia Institute of Technology, and Damien Hall and Thao Phuong Ngo from WPI-NanoLSI, Kanazawa University, for their valuable

contribution to the data analyses. The authors also acknowledge the financial support from KAKENHI (Japan Society for the Promotion of Science) for K.X.N. (19K06581, 23K05713, 23H02452-01) and T.Q.P.U. (#23H02452) and N.K. (20H00327); and CREST, Japan Science and Technology Agency (JPMJCR1762 to N.K.).

# Additional information

### Funding

| Funder | Grant reference number | Author |
| --- | --- | --- |
| Japan Society for the Promotion of Science | 19K06581 | Kien Xuan Ngo |
| Japan Society for the Promotion of Science | #23H02452 | Taro Uyeda |
| Japan Society for the Promotion of Science | 20H00327 | Noriyuki Kodera |
| Japan Science and Technology Agency | 10.52926/JPMJCR1762 | Noriyuki Kodera |
| Japan Society for the Promotion of Science | 23K05713 | Kien Xuan Ngo |
| Japan Society for the Promotion of Science | 23H02452-01 | Kien Xuan Ngo |

The funders had no role in study design, data collection and interpretation, or the decision to submit the work for publication.

### Author contributions

Kien Xuan Ngo, Conceptualization, Resources, Data curation, Formal analysis, Supervision, Funding acquisition, Validation, Investigation, Visualization, Methodology, Writing – original draft, Project administration, Writing – review and editing; Huong T Vu, Data curation, Formal analysis, Validation, Investigation, Methodology, Writing – original draft, Writing – review and editing; Kenichi Umeda, Software; Minh-Nhat Trinh, Formal analysis; Noriyuki Kodera, Visualization; Taro Uyeda, Formal analysis, Validation, Methodology, Writing – original draft, Writing – review and editing

### Author ORCIDs

Kien Xuan Ngo ⓘ https://orcid.org/0000-0002-9710-5452
Huong T Vu ⓘ https://orcid.org/0000-0002-1596-2507
Minh-Nhat Trinh ⓘ http://orcid.org/0000-0002-4154-5256
Noriyuki Kodera ⓘ http://orcid.org/0000-0003-4880-8423
Taro Uyeda ⓘ https://orcid.org/0000-0002-3584-9499

Reviewer #1 (Public Review): https://doi.org/10.7554/eLife.95257.3.sa1
Reviewer #2 (Public Review): https://doi.org/10.7554/eLife.95257.3.sa2
Author response https://doi.org/10.7554/eLife.95257.3.sa3

# Additional files

### Supplementary files

• MDAR checklist

### Data availability

The data generated or analyzed during this study are included in the manuscript. The source data is provided in Excel files attached to the appropriate figures.

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
