## [Editor Report · eLife assessment]

In this manuscript the authors present high-speed atomic force microscopy (HSAFM) to analyze real-time structural changes in actin filaments induced by cofilin binding. This **important** study enhances our understanding of actin dynamics which plays a crucial role in a broad spectrum of cellular activities based on **solid** experimental evidence. Some technical questions, however, remain, making the data interpretation **incomplete**.

---

## [Referee Report · Reviewer #1 (Public Review)]

The authors provided a detailed analysis of the real-time structural changes in actin filaments resulting from cofilin binding, using High-Speed Atomic Force Microscopy (HSAFM). The cofilin family controls the lifespan of actin filaments in cells by severing the filament and promoting depolymerization. Understanding the effects of cofilin on actin filament structure is critical. It is widely acknowledged that cofilin binding significantly shortens the pitch of the actin helix. The authors previously reported (1) that this shortening extends to the unbound region of the actin filament on the pointed end side of the cofilin binding cluster. In this study, the authors presented substantially improved AFM images and provide detailed accounts of the dynamics observed. It was found that a minimal cofilin-binding cluster, consisting of 2-4 molecules, could induce changes in the helical parameters over one or more actin crossover repeats. Adjacent to the cofilin-binding clusters, the actin crossovers were observed to shorten within seconds, and this shortening was limited to one side of the cluster. Additionally, the phosphate binding to the actin filament was observed to stabilize the helical twist, suggesting a mechanism in which cofilin preferentially binds to ADP-bound actin filaments. These findings significantly advance our understanding of actin filament dynamics which is essential for a wide of cellular processes.

However, two insufficient parts exist. Readers should be aware of possible errors in the Mean Axial Distance (MAD) analysis and the limitations of discussions about the actin subunit structure.

The authors have presented findings that the MAD within actin filaments exhibits a significant dependency on the helical twist. However, difficulty in determining each subunit interval from the AFM image might affect the analysis. For example, the observation of three peaks in HHP6 of Figure Supplement 6C, corresponding to 4.5 pairs, showed peak intervals of 5, 11.8, 8.7, and 5.7 nm (measured from the figure). The second region (11.8 nm) appears excessively long. If one peak is hidden in the second region, the MAD becomes 5.5 nm.

The authors also suggest a strong link between the C-form (cofilin binding form of actin found in cofilactin) and the formation of regions of the short pitch helix outside the cofilin binding cluster. However, the AFM observation did not provide any evidence about the actin form in these regions because of measurement limitations. Additionally, Oda et al. (2) have demonstrated that the C-form is highly unstable in the absence of cofilin binding, casting doubt on the possibility of the C-form propagating without cofilin binding. The "C-actin-like structure" in the paper is not necessarily related to the C-form actin. It might be one of the G-forms (monomeric actin forms) or another unknown form.

(1) K. X. Ngo et al., a, Cofilin-induced unidirectional cooperative conformational changes in actin filaments revealed by high-speed atomic force microscopy. eLife 4, (2015).

(2) T. Oda et al., Structural Polymorphism of Actin. Journal of molecular biology 431, 3217-3228 (2019).

---

## [Referee Report · Reviewer #2 (Public Review)]

Summary:

This study by Ngo et al. uses mostly high-speed AFM to estimate conformational changes within actin filaments, as they get decorated by cofilin. The authors build on their earlier study (Ngo et al. eLife 2015) where they used the same technique to monitor the expansion of cofilin clusters on actin filaments, and the propagation of the associated conformational changes in the filament (reduction of the helical pitch). Here, they propose a higher-resolution description of the binding of cofilin to actin filaments.

Strengths:

The high speed AFM technique used here is quite original to address this question, compared to more classical light and electron microscopy techniques. It can certainly bring valuable information as it provides a high spatial resolution while monitoring live events. Also, in this paper, a nice effort was made to make the 3D structures and conformational changes clear and understandable.

Weaknesses:

In spite of the authors' response to my earlier comments, I still have concerns regarding the AFM technique. In particular, regarding the interactions of the filaments with the surface, which I still find unclear and potentially problematic.

The filaments appear densely packed on the surface, and even clearly in register in some images (if not most images, e.g., Figs 3AD, 4BC, 5A, 8AC). I understand that there are practical reasons for this, but isn't there a risk that this could affect the result? Maybe I did not understand the authors' response well enough, but I did not see a clear control that would alleviate my concern.

The properties of the lipid layer and its interaction with the actin filaments are still unclear to me. A poor control of these interactions is a problem if one aims to measure conformational changes at high resolution. The strength of the interaction appears tuned by the ratio of lipids put on the surface to change its electrostatic charge. A strong attachment likely does more than suppress torsional motion (as claimed in Fig 8A). It may also hinder cofilin binding in several ways (lower availability of binding sites on the filament facing the surface, electrostatic interactions between cofilin and the surface, etc.). Here again, I was not fully reassured by the authors' response.

The identification of cofilactin regions relies on the additional height of the "peaks", due to the presence of cofilin. It thus seems that cofilin is detected every half helical pitch (HHP), and I still don't understand how the authors can make reliable claims regarding the presence or absence of cofilin between these peaks.

---

## [Author Response]

The following is the authors’ response to the original reviews.

**Reviewer #1 (Public Review):**
The authors provided a detailed analysis of the real-time structural changes in actin filaments resulting from cofilin binding, using High-Speed Atomic Force Microscopy (HSAFM). The cofilin family controls the lifespan of actin filaments in the cells by severing the filament and promoting depolymerization. Understanding the effects of cofilin on actin filament structure is critical. It is widely acknowledged that cofilin binding significantly shortens the pitch of the actin helix. The authors previously reported (1) that this shortening extends to the unbound region of the actin filament on the pointed end side of the cluster. In this study, the authors presented substantially improved AFM images and provide detailed accounts of the dynamics observed. It was found that a minimal cofilin-binding cluster, consisting of 2-4 molecules, could induce changes in the helical parameters over one or more actin crossover repeats. Adjacent to the cofilin-binding clusters, the actin crossovers were observed to shortened within seconds, and this shortening was limited to one side of the cluster. Additionally, the phosphate binding to the actin filament was observed to stabilize the helical twist, suggesting a mechanism in which cofilin preferentially binds to ADP-bound actin filaments. These findings significantly advance our understanding of actin filament dynamics which is essential for a wide of cellular processes.However, I propose that the sections about MAD and certain parts of the discussions need substantial revisions.

In this study, we leverage high spatiotemporal resolutions of high-speed atomic force microscopy (HS-AFM) to analyze real-time structural changes in actin filaments induced by cofilin binding. Furthermore, we experimentally demonstrate the inherent variability in twist conformations of bare actin filaments. Our study integrates HS-AFM with Principal Component Analysis (PCA) to elucidate the actin structure-dependent preferential cooperative binding of cofilin. We provide experimental evidence to substantiate a "proof of principle" regarding the flexible helical twists of actin filaments that regulate the functions of actin-binding proteins. This important study enhances our understanding of actin filaments’ dynamics and polymorphic structures which play crucial roles in a broad spectrum of cellular activities.

We appreciate the comments from Reviewer 1. Below, we address their concerns point by point.

MAD analysisThe authors have presented findings that the mean axial distance (MAD) within actin filaments exhibits a significant dependency on the helical twist, a conclusion not previously derived despite extensive analyses through electron microscopy (EM) and molecular dynamics (MD) simulations. Notably, the MAD values span from 4.5 nm (8.5 pairs per half helical pitch, HHP) to 6.5 nm (4.5 pairs/HHP) as depicted in Figure 3C. The inner domain (ID) of actin remains very similar across C, G, and F forms (2, 3), maintaining similar ID-ID interactions in both cofilactin and bare actin filaments, keeping the identical axial distance between subunits in the both states. This suggests that the ID is unlikely to undergo significant structural changes, even with fluctuations in the filament's twist, keeping the ID-ID interactions and the axial distances. The broad range of MAD values reported poses a challenge for explanation. A careful reassessment of the MAD analysis is recommended to ensure accuracy.

The central challenge to study “Protein Dynamics” in real time lies in bridging the gap in time scales: HS-AFM captures dynamics of proteins within the milliseconds to seconds range, whereas molecular dynamics (MD) simulations typically operate within the femtoseconds to microseconds domain. Protein dynamics encompass a spectrum of temporal scales, from atomic vibrations to molecular tumbling and collective motions in simulations. HS-AFM stands out as a potent technique for delving into protein dynamics, including processes like protein folding and conformational changes triggered by drugs or protein interactions. Additionally, a significant limitation of MD simulation is the spatial modeling constraint (~50 x 50 nm unit), which restricts the study of large complex biological systems. However, utilizing HS-AFM enables the construction of intricate protein models facilitating the real time imaging of their structures and dynamics during functional activity.

Regarding the suggestion about ID-ID interactions in both cofilactin and bare actin filaments, maintaining identical axial distances (ADs) between subunits in both states, our HS-AFM cannot provide atomic-level structural insights to address this issue. However, we demonstrate that the variability of OD twists in actin protomers could potentially lead to globally shorter half helical pitches (HHPs) and fewer protomer pairs per HHP (Figure 2, Figure supplement 2) (see lines 218-222). The fluctuation in filament’s twist is further supported by currently available experimental data, including our findings (Figure 3C) in this study (see our Discussion in lines 555-560).

The minimal change in local ID-ID interactions results in an unchanged global length of actin filaments in both cofilin-bound and unbound cases (Figure supplement 2). However, filament’s twists, as experimentally detected by EM, high-resolution interferometric scattering microscopy (iSCAT), HS-AFM, and in pseudo AFM, are changeable (see lines 555-560).

We have additionally reassessed the fluctuation and dynamics of MAD in F-ADP-actin and F-ADP.Pi-actin over time at high temporal resolution (Figure supplement 3, Video 3, Table supplement 5). These data are further explained in the Results section (lines 264-270).

Furthermore, we reassessed the broad range of MAD values in F-ADP-actin segments on both sides of large cofilin clusters over time (Figure supplement 8, Video 5). These findings are explained in the Results section (lines 333-337) and further discussed in the new results (lines 555-560).

In determining axial distances, the authors extracted measurements from filament line profiles. It is advised to account for potential anomalies such as missing peaks or pseudo peaks, which could arise from noise interference. An example includes the observation of three peaks in HHP6 of Figure Supplement 5C, corresponding to 4.5 pairs. Peak intervals measured from the graph were 5, 11.8, 8.7, and 5.7 nm. The second region (11.8 nm) appears excessively long. If one peak is hidden in the second region, the MAD becomes 5.5 nm.

We acknowledge the difficulty in identifying peaks within the regions of bare actin segments adjacent to cofilin clusters or within the cofilactin region. In the revised Figure supplement 6C (originally Figure supplement 5C), we did not assess peak intervals as suggested by Reviewer 1. The measurement of axial distance (AD) and the number of peaks within a HHP to calculate the correct MAD is further detailed in the Methods section (see HS-AFM data analysis and processing, highlighted in purple).

Additionally, the purpose of presenting these Figures supplement 6-7 is to directly compare the half helices and the number of protomer pairs per HHP between bare actin filaments and actin segments near the boundary between cofilactin and bare actin segments on the PE side in the same AFM images. In an original version of this paper, we have avoided including the MAD values measured in the cofilactin region (HHP6, HHP7) in Figure Supplement 7E, to mitigate the measurement errors.

Compiling histograms of axial distances (ADs) rather than focusing solely on MAD may provide deeper insights. If the AD is too long or too short, the authors should suspect the presence of missing peaks or pseudo-peaks due to noise. If 4.4 or 5.5 pairs/HHP regions tend to contain missing peaks and 7.5-8.5 pairs/HHP regions tend to contain pseudo peaks, this may explain the MAD dependency on the helical twist.

The measurement of axial distance (AD) and the number of peaks within a HHP to calculate the correct MAD is further detailed in the Methods section (see Analyses of pseudo AFM images of F-actin and C-actin structures constructed from existing PDB structures (e.g., Figure supplement 2); and HS-AFM data analysis and processing, highlighted in purple).

We disagree with Reviewer 1’s suggestion that compiling histograms of ADs, rather than focusing solely on MAD, may provide deeper insights. AFM imaging provides only a 2-dimensional (2D) surface structure, unlike the 3-dimensional (3D) structure offered by Cryo-EM. In AFM imaging, we cannot capture the object from different angles as Cryo-EM does. Therefore, AD values measured in 2D AFM images do not accurately represent the axial distance between two adjacent protomers along the same actin filament. Consequently, we relied on MAD values. Our results, including the fluctuation in the number of protomer pairs per HHP, are further supported by other studies (see our Discussion in lines 555-560).

Additionally, Figure 3E indicates a first decay constant of 0.14 seconds, substantially shorter than the frame rate (0.5 sec/frame). This suggests significant variations in line profiles between frames, attributable either to overly rapid dynamics or a low signal-to-noise ratio. Implementing running frame averages (of 2-3 frames) is recommended to distinguish between these scenarios. If the dynamics are indeed fast, the averaged frame's line profile may degrade, complicating peak identification. Conversely, if poor signal-to-noise ratio is the cause, averaging frames could facilitate peak detection. In the latter case, the authors can find the optimal number of frame averages and obtain better line profiles with fewer missing and pseudo-peaks.

We utilized state-of-the-art HS-AFM with high temporal and spatial resolution to capture the dynamic structures of F-ADP-actin and F-ADP.Pi-actin segments at higher frame rate of 0.2 sec/frame and 0.1 sec/frame, respectively (Figure supplement 3). As suggested, we implemented running frame averages (3 frames) in the ACF analyses. Consistently, our results indicate that the first time constant (t1) remains around 0.1-0.4 seconds, independent of the imaging rates (0.1 – 0.5 sec/frame), for AD between two adjacent actin protomers in F-actin bound with ADP or ADP.Pi (Table Supplement 5), and in the similar range of (t1), shown in Figure 3E. These significant experimental results support the notion that helical twists, the number of actin protomers per HHP, and MAD in bare F-actin segments, are intrinsically dynamic and fluctuate around the mean values over time (see further in lines 264-270; 333-337; and 555-560). It should be noted that our original ACF analyses did not include the averaging of running frames, thus eliminating the possibility of low signal/noise ratio in our analysis, as shown in Figure 3E-F.

DiscussionsThe authors suggest a strong link between the C-form of actin and the formation of a short pitch helix. However, Oda et al. (3) have demonstrated that the C-form is highly unstable in the absence of cofilin binding, casting doubt on the possibility of the C-form propagating without cofilin binding. Moreover, in one strand of the cofilactin, interactions between actin subunits are limited to those between the inner domains (ID-ID interactions), which are quite similar to the interactions observed in bare actin filaments. This similarity implies that ID-ID interactions alone are insufficient to determine the helical parameters, suggesting that the presence of cofilin is essential for the formation of the short pitch helix in the cofilactin filament. Thus, crossover repeats are not necessarily shortened even if the actin form is C-form.

We have experimentally observed a shortened bare half helix adjacent to cofilin clusters on the PE side at high spatial resolution, comprising fewer protomers than normal half helices. Thus, we hypothesized that crossover repeats are shortened if the actin protomers in the bare half helix neighboring the cofilin cluster on the PE side resembles a C-actin structure. This assumption is further explained by referring to C-actin structure in Figure 2 and Figure supplement 2. Even though the C-form, as suggested in Oda et al., 2019, is unstable, it intrinsically fluctuates around the mean value over time and adopts various conformations. A single PDB structure resolved by Cryo-EM through the ensembles of averaging structural images should be referenced as a single atomistic structure, one of many possible conformations, regardless it is resolved by Cryo-EM, X-ray diffraction or crystallography, or NMR (see Figure 1, legend of Figure supplement 1).

We highlight two main points regarding this issue: (1) The short helical pitch at the global scale is associated with the twisting of the OD at the local scale for individual protomers; (2) Actins in different nucleotide or cofilin bound states exhibit varying ranges, distributions, spectra, variations of both local OD twist and global helical pitch (Figure 1-2, Figure supplement 1-2). The first point underscores that the twist/untwist of the OD determines the shortness of the helical pitches, rather than the ID-ID interactions. The latter point is more related to the global length of the filament. The minimal change in local ID-ID interactions results in an unchanged global length of actin filaments in both cofilin-bound and unbound cases see pseudo AFM images in Figure supplement 2 for canonical actin filament and cofilactin segments with the same length (comprising 62 protomers). However, filament’s twists, as experimentally detected by EM, high-resolution interferometric scattering microscopy (iSCAT), HS-AFM, and in pseudo AFM, are changeable (see lines 555-560) and independent on the ID-ID interactions.

Narita (4) proposes that the facilitation of cofilin binding may occur through a shortening in the helix pitch, independent of a change to the C-form of actin. Furthermore, the dissociation of the D-loop from an adjacent actin subunit leads directly to the transition of actin to the G-form, which is considered the most stable configuration for the actin molecule (3).

See also our explanation above. We have incorporated these points in a Discussion section. See lines 497-499; 510-511.

Furthermore, our PCA analysis indicates that the transition from C-actin to G-actin necessitates the opening of the nucleotide cleft (resulting in a decrease in PC1) and is more readily achieved than the direct transition from F-actin to G-actin (which requires decreases in both PC1 and PC2). Whether this transition is directly triggered by the dissociation of the D-loop remains a topic for our future investigations. Our PCA analysis reveals that the D-loop is deeply buried within the core of the filament (Figure 2). Further experiments will be conducted to elucidate its roles.

The mechanism by which the shortened pitch propagates remains a critical and unresolved issue. It appears that this propagation is not a result of the C-form's propagation but likely involves an unidentified mechanism. Identifying and understanding this mechanism represents an essential direction for future research.

It's worth mentioning that our HS-AFM data and spatial ACF analysis lend support to a hypothesis suggesting that 2-4 bare actin protomers adjacent to cofilin clusters on the PE side adopt C-actin-like structures. Additionally, we have proposed several hypotheses aimed at better understanding the mechanisms driving the unidirectional binding and expansion of cofilin clusters toward the PE side. These hypotheses will require further examination in future experiments. Additional information can be found in lines 328-329; 344-351; and 416-430.

(1) K. X. Ngo et al., a, Cofilin-induced unidirectional cooperative conformational changes in actin filaments revealed by high-speed atomic force microscopy. eLife 4, (2015).(2) K. Tanaka et al., Structural basis for cofilin binding and actin filament disassembly. Nature communications 9, 1860 (2018).(3) T. Oda et al., Structural Polymorphism of Actin. Journal of molecular biology 431, 3217-3228 (2019).(4) A. Narita, ADF/cofilin regulation from a structural viewpoint. Journal of muscle research and cell motility 41, 141-151 (2020).

We have cited them accordingly in the paper.

**Reviewer #2 (Public Review):**
Summary:This study by Ngo et al. uses mostly high-speed AFM to estimate conformational changes within actin filaments, as they get decorated by cofilin. The authors build on their earlier study (Ngo et al. eLife 2015) where they used the same technique to monitor the expansion of cofilin clusters on actin filaments, and the propagation of the associated conformational changes in the filament (reduction of the helical pitch). Here, they propose a higher-resolution description of the binding of cofilin to actin filaments.Strengths:The high speed AFM technique used here is quite original to address this question, compared to classical light and electron microscopy techniques. It can certainly bring valuable information as it provides a high spatial resolution while monitoring live events. Also, in this paper, a nice effort was made to make the 3D structures and conformational changes clear and understandable.

We are grateful for the positive feedback from Reviewer 2.

Weaknesses:The paper also has a number of limitations, which I detail below.In addition to AFM, the authors also propose a Principal Component Analysis (PCA) of exisiting structural data on actin protomers. However, this part seems very similar to another published work by others (Oda et al. JMB 2019), which is not even cited.

We addressed this issue and explained it in Methods section, lines 612-621.

The asymmetrical growth of cofilin clusters has so far only been seen using AFM, by the same authors (Ngo et al. eLife 2015). Using fluorescent microscopy, others have reported a very symmetrical expansion of cofilin clusters (Wioland et al. Curr Biol 2017). This is not mentioned at all, here. It should be discussed, and explanations for this discrepancy could be proposed.

We have cited this paper (Wioland et al. Curr Biol 2017) in the current manuscript (see lines 361-362). However, we are unable to evaluate the technical distinctions between our methods and theirs. Instead, we have referred to a more recent paper that employed similar techniques to those used by Wioland et al. in Current Biology 2017. Our findings align with those reported by Bibeau JP et al. in the Journal of Molecular Biology 2021 see their Results on page 7, titled “Cofilin clusters elongate preferentially towards the actin filament pointed end”. At the minimum, we believe this is appropriate.

Regarding the AFM technique, I have the following concerns.The filaments appear densely packed on the surface, and even clearly in register in some images (if not most images, e.g., Figs 3A, 4BC, 5A). Why is that? Isn't there a risk that this could affect the result? This suggests there is some interaction between the filaments.

In this study, as well as in many similar studies of actin filaments alone or in interaction with other actin binding proteins (ABPs) including cofilin, we have carefully considered the density of filaments when designing experiments. We used highly dense, but not packed, actin filaments to minimize free space between filaments and the surface, which helps maintain stable tip-scanning during AFM imaging. This strategy technically allows us to capture high spatial and temporal resolutions of actin filaments’ structures.

The actin filaments, resemble paracrystal structures, are represented as densely packed actin filaments (see our data in Ngo and Kodera et al., eLife 2015, Figure 1C). Thus, the data presented in this paper is technically appropriate and does not risk misinterpretation due to lateral interactions impacting the structures and function of actin filaments and cofilin.

The properties of the lipid layer and its interaction with the actin filaments are not clear at all. A poor control of these interactions is a problem if one aims to measure conformational changes at high resolution. The strength of the interaction appears tuned by the ratio of lipids put on the surface to change its electrostatic charge. A strong attachement likely does more than suppress torsional motion (as claimed in Fig 8A). It may also hinder cofilin binding in several ways (lower availability of binding sites on the filament facing the surface, electrostatic interactions between cofilin and the surface, etc.)

We are confident that our lipid membrane bilayer is the optimal choice for immobilizing actin filaments in a controlled manner for HS-AFM experiments, achieved through the variation of positively charged lipids. In this study, we have fine-tuned the surface charge for our specific purposes.

As an example, to capture high-spatial resolution images of actin structures (Figure 5-6, Figure supplement 5B, 6), we strongly fixed the filaments on DPPC/DPTAP (50/50 wt%) after the binding reaction between actin filaments and cofilin in solution was completed. This experiment yielded valuable information, including: (i) the ability to replicate the conformation of cofilactin and hybrid cofilactin/bare actin segments in solution, akin to the first steps in sample preparation for Cryo-EM techniques; and (ii) the capability to capture these structures, reflecting their solution states, by firmly fixing them on a lipid surface. On the lipid surface, these structures were retained stably during AFM imaging.

If there is a choice, we advise against using amino-silane and other positively charged polymers typically used for modifying glass surfaces to fix actin filaments in studies using fluorescence microscopy. The strong immobilization by these chemicals can alter the structural dynamics and functions of actin filaments, lead to non-specific binding of cofilin on the modified glass surface, and potentially affect data interpretation.

On a local scale, the reviewer may argue about the "lower availability of binding sites on the filament facing the surface". However, on a global scale, we maintain that two single strands forming helical twists of long F-actin segments should have an equal chance to bind cofilin even when fixed on a lipid membrane. The evidence shown in Figure 8A and Video 7, which demonstrates that small cofilin clusters associate and dissociate locally without developing into large clusters along the actin filament, supports our conclusion that flexibility and dynamics in helical twists plays a crucial role in facilitating the binding and growth of cofilin clusters.

The lipid surface utilized in our study with actin filaments and cofilin provides an ideal surface, as it is flat and minimizes the nonspecific binding of cofilin to the lipid membrane (see an example of the lipid surface in Video 5).

How do we know that the variations over time are not mostly experimental noise, i.e. variations between repeats of the same measurement? As shown in Fig 3, correlation is mostly lost from one image to the next, and rather stable after that.

This question is similar to the above question of Reviewer 1. Please also refer to our response in lines 264-270; 333-337; 555-560, measurement Methods, and Figure supplement 3 and Table supplement 5.

The identification of cofilactin regions relies on the additional height of the "peaks", due to the presence of cofilin. It thus seems that cofilin is detected every half helical pitch (HHP), but not in between, thereby setting the resolution for the localization of cluster borders to one HHP. It thus seems difficult to claim that there is a change in helicity without cofilin decoration over this distance. In Fig 7, the change in helicity could be due to cofilin decoration that is undetected because cofilins have not yet reached the next peak.

There are several important criteria to distinguish the "supertwisted half helix" in cofilactin region from the "normal half helix". As illustrated in the pseudo AFM images constructed for normal F-actin and C-actin segments (with and without cofilin decoration) from PDB structures, it is evident that these two structures differ significantly in length and the number of protomer pairs per HHP (see Figure Supplement 2). In both pseudo and experimental AFM images, these parameters can be easily detected by measuring the distance between two cross-over points. Furthermore, the height or thickness difference between the cofilactin and bare actin regions is approximately 10-15 Å, which is well resolved by HS-AFM due to its exceptional z-axis resolution of ~1 Å. Technically, we were able to detect these differences by creating a longitudinal section profile that covered both bare actin and cofilactin areas, as shown in Figure supplement 6.

We experimentally reveal that a critical cofilin cluster comprising 2-4 molecules (Figures 5-6) or larger cofilin clusters (Figures 7-8, Figure Supplements 6-8) could equally supertwist a bare half helix on the PE side. The observation that a small cofilin cluster (2-4 molecules) can shorten a half helix by reducing number of protomers per HHP to 9 or 11 (4.5 or 5.5 protomer pairs), which typically requires full decoration by 9-11 cofilin molecules, strongly suggests that supertwisting or the change in helicity does not always require complete cofilin decoration. We predicted that 2-4 bare actin protomers neighboring a cofilin cluster on the PE side can adopt the C-actin-like structure. See further in lines 324-329.

Figure 7 captures a live binding event of cofilin at low spatial resolution, yet (i) the half helical pitches and (ii) the thickness of the cofilactin and bare actin segments can still be clearly distinguished. This demonstrates that changes in helicity within the cofilactin region propagate to an unbound half helix on the PE side, rearranging the helical twist by reducing the number of actin protomers per HHP, prior to recruiting additional cofilin for binding and expanding clusters.

**Reviewer #1 (Recommendations For The Authors):**
I believe C-form and G-form are better than C-actin like structure or G-actin like structure.

We avoid using terms like "G-form", "F-form", or "C-form", as defined by Cryo-EM (Oda et al., 2019), because they refer to specific nucleotide and cofilin-bound states in other original papers. Instead, we use “G-actin”, “F-actin”, “C-actin”, “G-actin-like”, and “C-actin-like” to emphasize "Structural Dynamics" and "Structural Polymorphism". This highlights that even F-actin structures without cofilin bound can adopt "C-actin-like" conformations with fewer OD twists, resulting in a shorter global helical pitch. ADP-bound F-actins exhibit greater variability in helical twists than ADP-Pi-bound F-actin (Figure 9), indicating that ADP-bound F-actin protomers can adopt more C-actin-like conformations than ADP-Pi-bound F-actin protomers (Figure 1, Figure supplement 1).

Technical terms describing actin structures do not need to be the same between Cryo-EM and HS-AFM, as the two techniques are fundamentally different. Our work underscores the importance of considering “structural dynamics and heterogeneity” in different nucleotide states of filamentous actin structures, both with and without cofilin, over time.

Figure 1AA very similar analysis has already been performed by Oda et al (1). The authors should describe the relationships with the previous analysis.

We addressed this issue in Methods – Principal component analysis – in lines 612-621.

Figure 1B, CA very similar analysis has already been performed by Tanaka et al. (2). The authors should describe the relationship with the previous analysis.

We addressed this issue in Methods – Principal component analysis – in lines 612-621 and legend of Figure 1.

Lines 397-398"However, we noted that in rare instances, cofilin clusters also grew on both sides in the regular bare half helices when ATP or ADP was present."I believe other experiments also contain ATP in the solution. I could not catch the meaning of this sentence.

We addressed this issue in the Results section, line 412. "However, we noted that in rare instances, cofilin clusters also grew on both sides in the regular bare half helices when only ADP was present."

Additionally, we enhanced the description in the Methods section to avoid any confusion regarding nucleotides in the buffer. Please refer to the Methods section under “HS-AFM imaging”, lines 702-738.

Lines 427-429"Consequently, the proportion of naturally supertwisted half helices with HHPs shorter than 30 nm was 5.8% for F-ADP-actin but only 1.1% and 0.2% for F-ADP.Pi-actin and phalloidin-stabilized F-actin, respectively."Similar discussion was made in (3) for the actin filaments with tension. It might be comparable with the current data.

We cited it accordingly, line 447 for Okura et al., 2023.

Lines 553-557"Nonetheless, it remains plausible that the structural flexibility exhibited 553 by ADP-bound actin protomers could result in subtle variations in the conformations of the DNase binding loop (Dloop) G46-M47-G48-N49, as suggested in (Chou and Pollard, 2019). We suggest that the absence of bound Pi possibly increases the torsional flexibilities during helical twisting of ADP bound actin filaments in contrast to their ADP.Pi-bound counterparts."The crystal structure of the F-form (4) showed that Pi in ADP.Pi connects the two large domains of the actin molecule, stabilizing F-form. Pi release largely weakens the connection. This might be useful for the discussion.

We incorporated this point with the suggested citation in lines 582-584.

(1) T. Oda et al., Structural Polymorphism of Actin. Journal of molecular biology 431, 3217-3228 (2019).

(2) K. Tanaka et al., Structural basis for cofilin binding and actin filament disassembly. Nature communications 9, 1860 (2018).

(3) K. Okura et al., Mechanical Stress Decreases the Amplitude of Twisting and Bending Fluctuations of Actin Filaments. Journal of molecular biology 435, 168295 (2023).

(4) Y. Kanematsu et al., Structures and mechanisms of actin ATP hydrolysis. Proceedings of the National Academy of Sciences of the United States of America 119, e2122641119 (2022).

**Reviewer #2 (Recommendations For The Authors):**
Line 190: "Noticeably, PCA analysis revealed higher structural flexibility in F-ADP-actin (red dots), exploring a larger space than F-ADP-Pi-actin structures (orange dots) within the F-actin cluster (inset in Figure 1A)". Is there a quantification to support this claim? Visually, things are not so clear.

We have improved Figure 1 by adding 2 circles to an inset, providing clearer quantification to support our claim.

In the PCA part: isn't it a bit obvious, or at least expected, that the conformation adopted by actin in the cofilactin structure is the most favorable one for binding cofilin?

We agree this point with the reviewer and have added this point accordingly in the Results section, lines 202-204.

I found it a bit unclear how the structures in Fig 2 were obtained.

We further explained it by adding “Zoom-in views of these long filaments are shown in Figure 2” in Methods section, line 661.

In the AFM images, the authors always seem to know the polarity of the filaments. Unless I missed it, how they know this is not explained. In their earlier work (Ngo et al. 2015) they used a subfragment of myosin II which indicates polarity when bound to F-actin. I found no such explanation here.

We have addressed this issue in the legend of each figure accordingly.

For clarity, I suggest writing "C-actin-like structures" (with two hyphens) rather than "C-actin like structures".

We agree and are currently incorporating this change in the text.

The term "cluster" in PCA can be confusing because it is used for cofilin clusters throughout the text.

"Cluster" is a common term used in PCA analysis. To clarify, we revised the legend in Figure 1 and Figure Supplement 1, changing "PCA clusters" to distinguish them from “cofilin clusters” or “F-actin clusters”.

There are many acronyms. Readibility of the figure legends (which can be consulted independently from the main text) would be improved if acronyms were explicited there as well.

We have revised some of the acronyms in the legend of each figure accordingly. At the minimum, we believe it is appropriate.